# Recycling Pretrained Classification Heads for Efficient Vision-Language Alignment

## Abstract

Vision-Language Models (VLMs) with separate image and text encoders, such as CLIP, excel at tasks like zero-shot classification or cross-modal retrieval. They achieve this by embedding images and text into a shared representation space. However, their success relies on end-to-end training with large volumes of paired samples, entailing prohibitive data and computational costs. Existing post-hoc vision-language alignment methods, which map independently trained image and text encoders into a shared representation space using lightweight functions, reduce training costs but still require substantial paired data. We introduce a data augmentation approach that recycles classification head weights from ImageNet-21K pretraining and combines them with a reduced number of image-text pairs to achieve vision-language alignment. These recycled weights significantly mitigate the need for large alignment datasets, while the combination with a reduced number of image-text pairs extends alignment beyond the original ImageNet domain. We demonstrate that integrating our augmentation approach with several state-of-the-art post-hoc alignment techniques consistently boosts accuracy in cross-modal retrieval, zero- and few-shot classification tasks. Experiments confirm that our approach provides a versatile and data-efficient solution for vision-language representation alignment.

## 1 Introduction

Vision-language models have unlocked powerful capabilities—such as zero-shot classification, cross-modal retrieval, image captioning, and visual question answering—by embedding images and text into a shared representation space. Classic approaches (Radford et al., 2021; Jia et al., 2021) achieve this by jointly training image and text encoders with massive paired datasets. However, collecting and curating billions of image-caption pairs, and then optimizing two large models end-to-end, incurs prohibitive data and computational costs for most researchers and institutions, effectively restricting the training of such models to those with considerable resources. Recent *decoupled* techniques (Zhai et al., 2022; Cao et al., 2025) freeze an encoder pretrained on one modality and then rely on contrastive training to align the other modality's encoder to this fixed representation, still requiring plenty of data and computation. Post-hoc alignment methods (Norelli et al., 2023; Li et al., 2025; Maniparambil et al., 2024) eliminate joint contrastive training but still rely on extensive paired image-text data to achieve vision-language alignment.

We propose combining the classification head weights of vision models pretrained on ImageNet-21K (Ridnik et al., 2021), typically discarded after pretraining, with image-text representations. In particular, we treat classification heads obtained during image feature extractor pretraining (e.g., ConvNeXt (Liu et al., 2022), ResNet (He et al., 2016), ViT (Dosovitskiy et al., 2020)) in the same way as image-text representation pairs (Figure 1) to learn an effective alignment with text encoders (e.g., BERT (Devlin et al., 2019), RoBERTa (Liu et al., 2019), MPNET (Song et al., 2020)). Our approach both reduces the amount of paired image-text data required and enables vision-language alignment across domains that differ from those used to train the classification head weights. We demonstrate that it boosts the accuracy of several state-of-the-art alignment methods (Li et al., 2025; Moayeri et al., 2023) in cross-modal retrieval and achieves promising results in zero- and few-shot classification. We highlight our key contributions here:

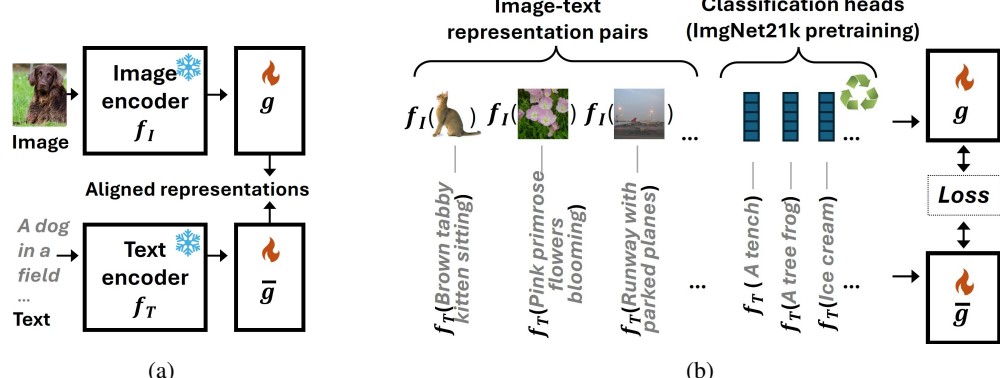

Figure 1: Overview of the proposed data augmentation approach for post-hoc representation alignment. (a) Illustration of the post-hoc alignment setting, in which lightweight functions $g$ and $\overline{g}$ are learnt to map representations from independently trained image and text encoder to an image-text aligned space. (b) Our approach, where we recycle the classification head weights from ImageNet-21K pretraining with their class names to augment representation data for learning $g$ and $\overline{g}$.

- To the best of our knowledge, we introduce the first approach that leverages pretrained classification heads, in combination with image-text data, to align independently trained image and text encoders.

- By recycling classification heads that are typically discarded, our method enables effective vision-language alignment with minimal image-text pair data, achieving strong performance in cross-modal retrieval, zero- and few-shot classification.

- We conduct comprehensive ablation studies to evaluate the benefits of ImageNet-21K versus ImageNet-1K classification heads and to assess how incorporating small amounts of image-text data improves domain-specific alignment. Additionally, we employ multimodal alignment metrics to compare how well classification head weights and image representations align with the text embeddings of their corresponding semantic concepts.

This work paves the way for rapid and resource-efficient vision-language alignment by leveraging *recycled* pretrained classification heads to augment paired image-text data.

## 2 RELATED WORK

**Vision-Language Models trained from scratch.** Pioneering vision-language models like CLIP (Radford et al., 2021) or ALIGN (Jia et al., 2021) jointly train vision and text encoders with a contrastive loss, so that an image and a text are mapped to similar representations if and only if they share the same semantic content. The resulting aligned vision-language space enables zero-shot transfer across downstream tasks without requiring task-specific fine-tuning. More recent approaches demonstrate that powerful off-the-shelf encoders can be leveraged to avoid training both modality encoders. LiT (Zhai et al., 2022) freezes a pretrained vision backbone and trains only the text encoder, while FLAME (Cao et al., 2025) does the opposite–freezing the text model and learning the parameters of the vision encoder. However, these approaches still require substantial computational costs, as they involve training a large encoder with hundreds of millions of parameters.

**Post-hoc vision-language alignment with image-text pairs.** A growing body of work focuses on connecting image and text encoders that were not jointly trained. The text-to-concepts framework (Moayeri et al., 2023), for example, starts with a pretrained vision-only encoder and learns a single-layer MLP that maps its image representations into CLIP's latent space. The combination of the vision encoder and the MLP, together with CLIP's text encoder, produces similar representations for images and texts that share the same semantic content, in a manner analogous to CLIP's original image and text encoders. Additionally, some methods aim to align image and text encoders that have been trained each exclusively on unimodal data. The seminal work ASIF (Norelli et al., 2023)

embeds visual and text data into a common space using a fixed set of image-caption pairs: for a given image, its representation is the vector whose $i$-th component is the similarity (in the original image encoder's space) between that image and the $i$-th image in the fixed set; likewise, for a given text, its representation is the vector whose $i$-th component is the similarity (in the original text encoder's space) between that text and the $i$-th caption in the fixed set. CSA (Li et al., 2025) applies Canonical Correlation Analysis to project pretrained image and text embeddings into a maximal correlation subspace. SAIL (Zhang et al., 2025) contrastively trains two linear transformations on millions of image-caption pairs to align representations from both modalities. The QPA method (Maniparambil et al., 2024) addresses alignment as a seeded graph-matching problem: it formulates a Fast Quadratic Assignment Problem optimization that directly aligns the similarity graphs of the two modalities. The issue with these approaches is that they require a sufficiently large and diverse set of multimodal image-text pairs for the alignment to work effectively, which may not be available in many practical domains and thus limits their applicability.

**Zero-shot classification by mapping text to weights.** A related line of work creates zero-shot classifiers by mapping text representations directly to the weights of image classifiers. For example, methods like ICIS (Christensen et al., 2023) and Zero-Shot Natural Language Explanations (Sammani & Deligiannis, 2025) use MLPs to transform text embeddings into classification weights. However, these approaches are constrained to the ImageNet-1K or other more specific domains, limiting their effectiveness beyond these settings.

While previous work focuses on either text-to-image alignment or text-to-weights mapping, with the latter constrained to ImageNet-1K or domain-specific datasets, we show that considering both simultaneously provides substantial gains. We leverage the larger-scale classification head weights from ImageNet-21K pretraining—despite being noisier than other pretraining sets such as ImageNet-1K—in combination with image-text pairs. This approach provides much wider domain coverage for alignment and can be flexibly extended to specific domains non-overlapping with ImageNet-21K by selecting appropriate image-text paired data. We note that the goal of our post-hoc alignment method is to minimize the paired data needed for the alignment step itself by leveraging high-quality representations provided by classifier weights. This is distinct from the (potentially large) datasets used to pre-train the image and text encoder components, which we treat as fixed, off-the-shelf models.

## 3 METHODOLOGY

Our approach leverages classification heads from ImageNet-21K pretraining alongside image-text pairs for vision-language alignment. This combination significantly improves the effectiveness of existing post-hoc alignment methods. We first start by formulating the post-hoc alignment problem.

**General post-hoc alignment setting.** By post-hoc alignment, we refer to methods that learn lightweight functions to align independently pretrained encoders, without modifying the encoder parameters. We formulate this task mathematically. Let $\mathcal{X}$ denote the input image space and $\mathcal{T}$ denote the input text space. We define our image and text encoders as:

$$f_I : \mathcal{X} \longrightarrow \mathbb{R}^d \qquad \text{and} \qquad f_T : \mathcal{T} \longrightarrow \mathbb{R}^{\overline{d}},$$

where $f_I$ maps each image into a $d$-dimensional feature vector and $f_T$ maps text inputs into $\overline{d}$-dimensional feature vectors. If $f_I$ and $f_T$ had been jointly optimized with a contrastive loss as in CLIP (Radford et al., 2021), then $d = \overline{d}$ and they would satisfy

$$f_I(x) \approx f_T(t)$$

for image-text pairs $(x, t)$ with the same semantic content (i.e. an image and a textual description of it), while pushing apart pairs with different semantic content. In the post-hoc alignment setting, we assume independently pretrained encoders, thus $f_I$ and $f_T$ are not aligned. To bridge this gap, the objective is that lightweight functions $g : \mathbb{R}^d \to \mathbb{R}^s$ and $\overline{g} : \mathbb{R}^{\overline{d}} \to \mathbb{R}^s$ are learned to map each modality's space into a common space $\mathbb{R}^s$ of dimension $s$, such that for any image-text pair $(x, t)$ sharing the same semantic content,

$$g(f_I(x)) \approx \overline{g}(f_T(t)) \tag{1}$$

To learn $g$ and $\overline{g}$, an alignment dataset

$$\widetilde{\mathcal{D}}_{imgtxt} = \{(x_j, t_j)\}_{j=1}^p \subseteq \mathcal{X} \times \mathcal{T}$$

with $p$ image-text pairs is used. Since $g$ and $\overline{g}$ act on the image and text representation spaces, we only need the representations of image-text pairs in $\widetilde{\mathcal{D}}_{imgtxt}$ to learn $g$ and $\overline{g}$. Thus, we define a dataset $\mathcal{D}_{imgtxt}$ as

$$\mathcal{D}_{imgtxt} = \{(f_I(x_j), f_T(t_j)\}_{j=1}^p$$

and focus on it instead of on the original image-text alignment dataset $\widetilde{\mathcal{D}}_{imgtxt}$. In the case of CSA (Li et al., 2025) and SAIL (Zhang et al., 2025), $g$ and $\overline{g}$ are linear projections learned using Canonical Correlation analysis and contrastive learning respectively. ASIF (Norelli et al., 2023) uses a fixed set $\mathcal{D}_{imgtxt}$ of image-text representations and defines $g$ as the function taking an image representation to a vector of similarities with respect to the vision part of $\mathcal{D}_{imgtxt}$ and analogous with $\overline{g}$ and text representations. In contrast to ASIF and CSA, the text-to-concepts approach (Moayeri et al., 2023) takes a different strategy. Instead of learning two lightweight mappings that project both modalities into a third representation space, it learns either $\overline{g}$ or $g$ to project one modality's representation space into the other modality's representation.

**Combining classification head weights and image-text representations.** Our approach recycles the classification heads of models pretrained on ImageNet-21K, to augment the $\mathcal{D}_{imgtxt}$ dataset and improve the effectiveness of different alignment methods that learn the functions $g$ and $\overline{g}$ (e.g., CSA, text-to-concepts). First, we assume the image encoder $f_I$ undergoes a pretraining phase, in which a classifier is built on top of $f_I$ as

$$W\, f_I(x) + b \tag{2}$$

where $W \in \mathbb{R}^{C \times d}$, with $C$ denoting the number of classes in the pretraining set (e.g., 21.841 in ImageNet-21K), and $b \in \mathbb{R}^C$ is the bias term of the linear layer. During pretraining, $W, b$ and the parameters of $f_I$ are optimized; afterwards, $f_I$ can be frozen as a fixed feature extractor. In this setting, our approach augments $\mathcal{D}_{imgtxt}$ with pairs of the form $(w_i, f_T(t_i))$ where $w_i$ is the i-th row of $W$ and $t_i$ the name of the i-th class in the pretraining dataset (e.g. 'A tench', 'A tree frog'). Formally, we define

$$\mathcal{D}_{weights} = \{(w_i, f_T(t_i))\}_{i=1}^C$$

and the augmented alignment dataset as

$$\mathcal{D}_{aug} = \mathcal{D}_{imgtxt} \bigcup \mathcal{D}_{weights} \tag{3}$$

which is then used it to learn $g$ and $\overline{g}$.

**Why treating classification head weights as representations?** In other words, what justifies the use of $(w_i, f_T(t_i))$ pairs to augment $(f_I(x_j), f_T(t_j))$ pairs within the same alignment dataset $\mathcal{D}_{aug}$? Our rationale is that row vectors $w_i$ from classification head weight matrix $W$ serve as effective *prototypes* of their corresponding semantic concepts. In our experiments in Section 4, we will empirically show that a cosine-similarity classifier of the form

$$\arg\max_i\ \cos(w_i, z) \tag{4}$$

achieves high classification accuracy. Since Equation (4) follows the form of prototype-based classifiers where decisions are made by comparing inputs to representative class vectors, the high classification accuracy implies that each $w_i$ encodes the essential characteristics of its associated classes in the image representation space. Hence, since $w_i$ effectively represents the semantic concept of class $i$, $(w_i, f_T(t_i))$ pairs can be used to augment $(f_I(x), f_T(t))$ pairs when learning the alignment functions $g$ and $\overline{g}$. Additional details are provided in Section 4.

**Why the classification heads from ImageNet-21K pretraining?** ImageNet-1K and ImageNet-21K are among the most widely employed publicly available labeled datasets for transfer learning (Ridnik et al., 2021). While other datasets such as Places365 or iNaturalist have also been used for transfer learning (Plested & Gedeon, 2022), they are less general and provide far fewer off-the-shelf pretrained checkpoints compared to the ImageNet benchmarks. ImageNet-1K dataset offers a clean and balanced structure: all classes contain the same number of samples, and each class corresponds to a leaf synset in WordNet—specific nouns with no hyponyms in WordNet ontology

(e.g., "chihuahua" but not "mammal")—ensuring semantic disjointness. By contrast, ImageNet-21K has an imbalanced class distribution and includes both leaf and non-leaf synsets (e.g., "chihuahua" and "mammal"), which introduces noise into the classification task from which the classification head weights are learnt. Nonetheless, ImageNet-1K's clean, uniform structure comes at the cost of limited concept diversity. Instead, ImageNet-21K, with approximately 21 times more weights, provides a much richer set of semantic representations for vision-language alignment. As we will demonstrate later, this increased scale compensates for the possible noise caused by overlapping categories and class imbalance.

## 4 EXPERIMENTS

Our main objective is to demonstrate that using the classification head weights from ImageNet-21K pretraining $D_{weights}$ to augment paired image-text representations $D_{imgtxt}$ significantly improves the effectiveness of different vision-language alignment methods. In particular, we will evaluate post-hoc aligned vision-language models across cross-modal retrieval, zero- and few-shot classification tasks. Lastly, we analyze how the classification head weights from ImageNet-21K pretraining can serve as faithful representations of semantic concepts, and compare them to image-based representations. Throughout this section, we use the BEiT-B16 (Bao et al., 2021) image encoder and CLIP's ViT-B/32 text encoder (Radford et al., 2021) as our image and text encoder respectively. Further experiments on tasks using additional image and text encoders—also those trained solely on text, such as RoBERTa (Liu et al., 2019) or MPNET (Song et al., 2020)—along with technical details (e.g., hyperparameters, hardware, training time) and code are provided in the Appendix.

**Cross-modal retrieval.** In our experiments on FLICKR30K retrieval (Plummer et al., 2015), we apply the augmentation defined in Equation (3) to three post-hoc alignment techniques—originally based only on image-text pairs—using ImageNet-21K classification head weights. Additional experiments on COCO (Lin et al., 2014) are provided in Appendix D. In addition to two state-of-the-art post-hoc alignment methods, such as CSA and text-to-concepts, operating in a low data regime, we employ a strong baseline that trains a lightweight MLP to map text representations to image representations, which we refer to as MLP-alignment. This approach—using a lightweight MLP trained with image-text pairs to align vision and language encoders—has been widely adopted in the literature, dating back to early image-text alignment works such as Frome et al. (2013). In particular, we choose a two-layer MLP, as this proved to yield better results than a single-layer MLP, and train it with cosine-similarity loss. For the three alignment techniques, we vary the amount of FLICKR30K training samples (0 to 30 K pairs), then evaluate the aligned image and text encoders in the cross-modal retrieval task on the FLICKR30K test set. For a fixed image $x'$, the retrieved text is:

$$\arg\max_{t \in T} \left\{ \cos\big(g(f_I(x')), \overline{g}(f_T(t))\big) \right\} \tag{5}$$

where $T$ is the set of possible texts (all the texts in the FLICKR30K test set), $g$ and $\overline{g}$ are the lightweight alignment functions obtained through the specific post-hoc alignment method (e.g., in MLP-alignment, $g$ is the identity function and $\overline{g}$ the trained MLP). Retrieval of an image given a

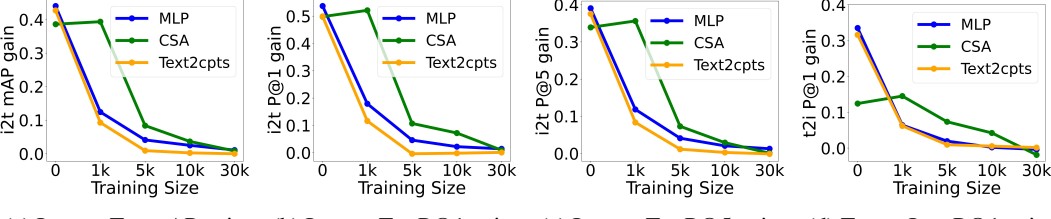

(a) Img-to-Txt mAP gain    (b) Img-to-Txt P@1 gain    (c) Img-to-Txt P@5 gain    (d) Txt-to-Img P@1 gain

Figure 2: Gains in FLICKR30K retrieval when augmenting image-text representations $D_{imgtxt}$ with classification head weights from ImageNet-21K pretraining $D_{weights}$. The x-axis shows the size of $D_{imgtxt}$. Post-hoc alignment techniques include CSA (Li et al., 2025), Text-to-Concept (Moayeri et al., 2023) and MLP alignment. Results show that augmenting with $\mathcal{D}_{weights}$ provides the largest gains in the low-data regime, with benefits diminishing as $\mathcal{D}_{imgtxt}$ grows—an expected outcome.

fixed text $t'$ is done in an analogous way. Notably, by leveraging classification head weights, our approach enables obtaining alignment functions $g$ and $\bar{g}$ even in the complete absence of paired image-text data, a scenario in which originally CSA, text-to-concepts and MLP-alignment could not operate. In Figure 2 we show the gains in retrieval metrics compared to using only image-text pairs for obtaining $g$ and $\bar{g}$. Complete results including absolute metric values (not only metric improvements) are provided in Appendix D. As more training image-text pairs from FLICKR30K are used in $\mathcal{D}_{imgtxt}$, the retrieval gains from augmenting the training set with classification head weights steadily diminish, tending towards zero when the full $\sim$30 K samples are used. This implies that our data augmentation approach delivers the largest benefits in the low-data regime, with gains that diminish as more image-text pairs become available.

A key question is whether improved retrieval stems merely from additional alignment data. We first note our method uses classification weight representations at no extra cost–they are pre-training by-products requiring no new image-text pairs. Secondly, we conducted an ablation study (Appendix D) comparing downstream retrieval performance when alignment uses weight representations versus an equivalent number of image-text pairs. Results show classification weights outperform image-text pairs, despite the latter matching the downstream task's data modality. This shows the superior quality of weight representations for alignment, even beyond their original classification purpose.

**Zero-shot classification.**    In the following experiment, we construct a zero-shot classifier by aligning an image and a text encoder with an MLP that maps text representations into image representations. The resulting zero-shot classifier demonstrates competitive capabilities even when the MLP is trained using the classification head weights from ImageNet-21K pretraining, with little to no additional image-text paired data required. In particular, we evaluate zero-shot performance of aligned vision and text encoders under different training configurations to address three objectives: first, analyzing how classification head weights from ImageNet-21K pretraining ($\mathcal{D}_{weights}$) improve zero-shot accuracy compared to ImageNet-1K classification heads; second, examining how incorporating even minimal image-text data ($\mathcal{D}_{imgtxt}$) during MLP aligner training significantly boosts performance compared to using $\mathcal{D}_{weights}$ alone; and third, demonstrating that the vision-only BEiT-B/16 encoder, when aligned to text with an MLP trained on $\mathcal{D}_{weights}$ and minimal image-text pairs $\mathcal{D}_{imgtxt}$, can compete in some datasets with the equivalent CLIP model (ViT-B/16). This competitive performance is achieved with only two minutes of training per MLP on a GTX Titan Xp.

Classification is performed by encoding the prompts "*A photo of <class i>*", and predicting image $x$ as the class whose representation has the highest cosine similarity with the image representation

$$\arg\max_{1 \le i \le C} \left\{ \cos\big(f_I(x), \mathrm{MLP}(f_T(\text{"A photo of <class i>"}))\big) \right\} \tag{6}$$

where MLP is being applied to the output of the text encoder $f_T$. Looking at Equation (1), $\bar{g}$ is the MLP and $g$ the identity, as we are mapping text to image representations. We evaluate our aligned vision-language model in zero-shot transfer across nine diverse datasets: RESISC45 (Cheng

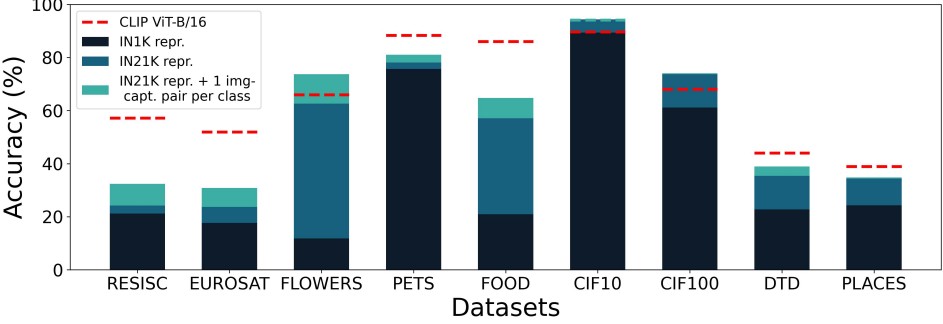

Figure 3: Zero-shot classification accuracy of BEiT-B/16 (Bao et al., 2021) aligned to text with an MLP. Stacked bars show the progressive accuracy gains: base performance when only classification head weights from ImageNet-1K pretraining are used during the MLP aligner training, additional gain from using ImageNet-21K weights, and further improvement from incorporating one image-caption pair per class, for all nine datasets at once. Red dashed lines show CLIP ViT-B/16 accuracy.

et al., 2017), EuroSAT (Helber et al., 2019), Flowers102 (Nilsback & Zisserman, 2008), OxfordPets (Parkhi et al., 2012), Food101 (Bossard et al., 2014), CIFAR-10, CIFAR-100 (Krizhevsky, 2009), DTD (Cimpoi et al., 2014), and Places365(Zhou et al., 2018). The different MLP training configurations for which we evaluate the resulting zero-shot classifier are the following:

- **ImageNet-1K**. $\mathcal{D}_{aug} = \mathcal{D}_{weights}$ where $\mathcal{D}_{weights}$ are formed by $(w_i, f_T(t_i))$ where $w_i$ is the i-th row of the $W$ matrix in the ImageNet-1K classification layer Equation (2) and $f_T(t_i)$ the text representation of this class.

- **ImageNet-21K**: Same setting as before but we use the $W$ matrix from the classification layer of an ImageNet-21K task. Despite being noisier due to class hierarchy complexity, classification head weights from the ImageNet-21K pretraining provide representations for significantly more classes than those from ImageNet-1K.

- **ImageNet-21K & 1 Image-Caption pair per class**: $\mathcal{D}_{aug} = \mathcal{D}_{imgtxt} \bigcup \mathcal{D}_{weights}$ where $\mathcal{D}_{weights}$ come from the ImageNet-21K classification head, as in the previous setting, but we add as $\mathcal{D}_{imgtxt}$ the pairs $(f_I(x), f_T(t))$, where one image $x$ is sampled from each class in each dataset and paired with an (LLM-generated) caption $t$. We note that we train a single MLP and then test zero-shot accuracy in all datasets. Since the total sum of classes across all nine datasets is $817$, the ratio between the size of $\mathcal{D}_{imgtxt}$ and the total alignment dataset $\mathcal{D}_{aug}$ is $\frac{817}{817+21,841} \approx 0.036$.

Results in Figure 3 show that transitioning from ImageNet-1K to ImageNet-21K classification head weights in $\mathcal{D}_{weights}$ yields consistent zero-shot classification improvements across all datasets. This suggests the $21\times$ larger set of representations provided by ImageNet-21K classification heads outweighs any potential noise from class overlap during MLP-aligner training. Secondly, adding image-text representation pairs $\mathcal{D}_{imgtxt}$ to the ImageNet-21K weight dataset $\mathcal{D}_{weights}$ further improves performance, but the gains vary by task dataset. Datasets with high class overlap with ImageNet-21K (e.g., CIFAR-10, CIFAR-100) show minimal improvements when including text-image pairs. This suggests that image-caption information provides diminishing returns when visual categories are already well-represented in the weight dataset. Finally, comparing the zero-shot classification accuracy of the aligned BEiT-B/16 model (Bao et al., 2021) with CLIP ViT-B/16, we observed that BEiT-B/16 achieves competitive accuracy on some datasets (OxfordPets, DTD, Places365) and even surpasses it on others (Flowers102, CIFAR-10, CIFAR-100). This strong performance of the BEiT-B/16 as a zero-shot classifier is achieved in a resource-efficient manner, both in terms of data and computation. With respect to data efficiency, by augmenting with the weight representations from ImageNet-21K pretraining we use only 817 image-caption pairs, while computationally our MLP requires only two GPU-minutes of training compared to CLIP's requirement of 400M image-text pairs and thousands of GPU hours (Radford et al., 2021).

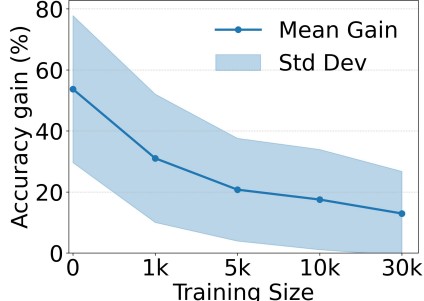

While ImageNet-21k pretraining weight representations may contain semantic overlap with downstream benchmarks, we note that CLIP has also likely seen most of the classes in the downstream classification benchmarks (Xu et al., 2024). For instance, Xu et al. (2024) reconstructs CLIP's data curation process and finds over 700 out of the 1K classes in ImageNet-1K present in pretraining metadata and observes a correlation between downstream zero-shot classification accuracy and the number of classes matched in the metadata. We can also infer from Xu et al. (2024) that our use of 1 image-caption pair per class across all datasets is likely less than the number of overlapping image-caption pairs CLIP saw during its pretraining phase.

Figure 4: Averaged zero-shot accuracy gains when augmenting alignment data with recycled ImageNet-21K weights.

Finally, we evaluate the zero-shot classification gains from augmenting image-text pairs with weight representations in Figure 4. The figure shows the mean zero-shot accuracy improvement across nine classification datasets when augmenting Flickr30k alignment data with recycled ImageNet-21K weights. Using the MLP alignment strategy, we observe consistent performance gains, particularly in low-data regimes. While we present MLP results here for clarity, other alignment methods (CSA and Text2Concepts) show similar trends, as detailed in Figure 7 from Appendix E.

Table 1: Performance comparison across few-shot methods and number of shots per class $K$. Experiments were conducted using five random seeds, and results are reported as mean $\pm$ standard deviation. Best results with statistical significance are in bold.

| K | Method | RESISC | EUROSAT | FLOWERS | PETS | FOOD | CFR10 | CFR100 | DTD | PLACES |
|---|--------|--------|---------|---------|------|------|-------|--------|-----|--------|
| 1 | Ours | 43.51±1.28 | 56.65±4.06 | 95.59±0.65 | **84.20**±0.88 | **67.53**±0.59 | **93.34**±0.27 | **75.14**±0.25 | 47.36±1.34 | **35.51%**±0.22 |
|   | NCC | 39.27±3.50 | 56.38±2.61 | 94.49±0.75 | 76.09±3.63 | 51.71±0.92 | 76.03±3.95 | 49.98±0.96 | 42.01±2.69 | 21.96%±0.71 |
|   | KNN | 39.27±3.50 | 56.38±2.61 | 94.49±0.75 | 76.09±3.63 | 51.71±0.92 | 76.03±3.95 | 49.98±0.96 | 42.01±2.69 | 21.96%±0.71 |
| 2 | Ours | 52.14±0.95 | 64.59±3.82 | 97.41±0.70 | **85.98**±0.51 | **70.96**±0.61 | **93.86**±0.49 | **75.77**±0.09 | 53.73±1.72 | **35.94%**±0.22 |
|   | NCC | 53.49±3.46 | 66.36±2.63 | 97.64±0.73 | 82.89±2.16 | 63.47±0.84 | 86.83±1.66 | 61.64±0.86 | 52.76±1.73 | 30.22%±0.51 |
|   | KNN | 38.77±2.89 | 52.17±3.08 | 93.69±0.35 | 73.99±0.98 | 50.07±1.82 | 71.67±4.65 | 48.50±0.80 | 41.58±3.04 | 22.49%±0.38 |
| 4 | Ours | 58.54±0.74 | 75.66±2.41 | 98.80±0.20 | 89.60±0.77 | 74.41±0.61 | **94.81**±0.36 | **76.42**±0.22 | 62.04±0.95 | **41.98%**±0.35 |
|   | NCC | **64.53**±0.97 | 76.27±2.50 | 98.91±0.34 | 88.17±1.78 | 73.60±0.83 | 91.01±1.40 | 71.35±0.60 | 61.54±1.13 | 38.71%±0.36 |
|   | KNN | 51.26±1.13 | 67.25±2.36 | 97.08±0.38 | 82.73±1.67 | 63.49±0.89 | 84.69±1.15 | 62.41±0.56 | 51.64±1.51 | 28.77%±0.34 |

**Few-shot classification.** Beyond cross-modal retrieval and zero-shot classification, our approach can be used for few-shot classification in the following way:

1. Initial alignment: Train a lightweight MLP (the same used in the retrieval and zero-shot setting) to map text representations to image representations using only the ImageNet-21K classification head (no image–text pairs).

2. Fine-tuning on image–text data: Further fine-tune this MLP on $\mathcal{D}_{imgtxt}$, which consists of image–text pairs from the target few-shot dataset

$$\big(f_I(x),\ f_T(\text{“A photo of }\langle\text{class}\rangle\text{”})\big)$$

We employ sequential rather than joint training on the combined dataset $\mathcal{D}_{aug} = \mathcal{D}_{imgtxt} \bigcup \mathcal{D}_{weights}$ to make the alignment more focused on the specific classification domain of interest. Classification follows the same protocol as zero-shot evaluation using Equation (6). We evaluate our approach against two strong baselines on frozen backbones: the Nearest Centroid Classifier (NCC) and K-Nearest Neighbors (KNN). The NCC, which computes class centroids from training representations and assigns test images to the nearest centroid, has been shown to be remarkably effective for few-shot classification, often surpassing more complex meta-learning approaches (Luo et al., 2023). We test across nine different classification tasks and standard few-shot settings with 1, 2 and 4 shots per class. Results in Table 1 show that our classifier performs significantly better than NCC and KNN in most cases. We acknowledge that a classifier based on Equation (6) with simple text prompts (*A photo of <class>*) may be suboptimal for few-shot scenarios. Future work could explore prompt tuning or adapter-based approaches with the aligned vision-language model. However, our results demonstrate that even this straightforward approach surpasses strong baselines like NCC, validating the effectiveness of our weight recycling strategy for few-shot learning.

**Classification head weights row vectors as prototypes.** Our approach relies on the assumption that each row $w_i$ of the weight matrix $W$ in Equation (2) can be interpreted as a prototype of its associated label in the image representation space. To validate this assumption, we extract the $w_i$ vectors corresponding to the ImageNet-1K classes from a model trained on ImageNet-21K, and compare two classifiers on the ImageNet-1K validation set: (i) the standard linear classifier, and (ii) a cosine-similarity classifier using $\cos(w_i, f_I(x))$. We conduct this experiment across five vision architectures covering transformer (BEiT, TinyViT), convolutional (ConvNeXt), and hybrid designs (CAFormer, ConvFormer). As shown in Table 2, the cosine classifier consistently achieves high accuracy (above 80% across all architectures). This suggests that the weight vectors $w_i$ encode meaningful representations of the concepts associated to the corresponding classes.

Table 2: Accuracy (%) on ImageNet-1K validation set for a linear layer $W f_I(x) + b$ and cosine similarity $\cos(w^i, f_I(x))$, where $w_i$ is $W$'s i-th row. The high accuracy in the cosine classifier indicates the $w_i$s are meaningful representations of their classes, for all encoders.

| Last layer | BEiT-B/16 | CAFormer-S18 | ConvFormer-S18 | ConvNeXt-B | TinyViT-21M |
|------------|-----------|--------------|----------------|------------|-------------|
| $W f_I(x) + b$ | 82.71 | 80.81 | 80.76 | 83.29 | 82.31 |
| $\cos(w_i, f_I(x))$ | 82.10 | 80.28 | 80.08 | 82.67 | 81.30 |

The Neural Collapse phenomenon (Papyan et al., 2020) could be argued to provide theoretical insight into why classifier weight rows can serve as class prototypes. This phenomenon describes the convergence of neural network parameters to a geometric configuration with self-duality: the row vector $w_i$ of $W$ and the mean of class $i$ representations converge up to rescaling. In the training limit, $w_i$ represents a (scaled) mean of image representations (see Appendix G for theoretical discussion), making it natural to use pairs $(w_i, f_T(\text{"A photo of class } i\text{"}))$ to augment the $(f_T(x), f_T(t))$ image-text representations. However, Neural Collapse has been studied under terminal phase training with zero-error, number of classes not exceeding feature dimension, and balanced class distributions. While recent extensions address some of these limitations individually by adding new technical conditions (Jiang et al., 2024; Yang et al., 2022), to the best of our knowledge, there are still gaps between theoretical assumptions and our ImageNet-21K pretraining setting. Hence, we examined the cosine classifier accuracy Table 2 as direct evidence that the $w_i$ vectors encode meaningful class representations, even when the conditions for Neural Collapse are not fully satisfied.

**Comparison of Semantic Alignment: Head weights vs Averaged Image Representations.** We now investigate how well our weight-derived prototypes align with text embeddings relative to conventional image representations. To determine whether the row vectors of the classification head weight matrix or the averaged image representations align more closely with text representations, we employ the mutual k-nearest-neighbor alignment metric ($m_{NN}$) (Huh et al., 2024). This metric measures how well two spaces preserve semantic neighborhoods. We extract the 1,000 row vectors $w_i$s from the ImageNet-21K classification head matrix that correspond to ImageNet-1K classes and compute their $m_{NN}$ alignment scores with the text embeddings of the corresponding class names. In parallel, we compute $m_{NN}$ scores for averaged image embeddings from the ImageNet-1K validation set, where we

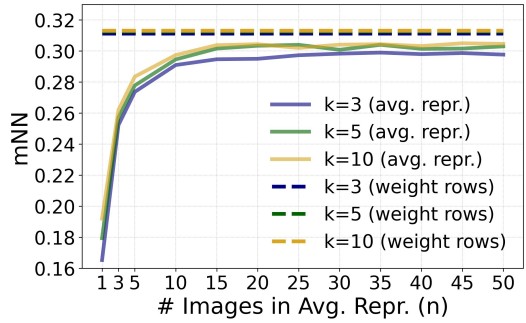

Figure 5: Mutual k-NN alignment ($m_{NN}$) to text representations for ImageNet-21K classification head vectors (dashed) and averaged image embeddings (solid) as a function of the number of images per class $n$. Multiple neighbors $k \in \{3, 5, 10\}$ are tested. Across all tested $k$ and $n$, classification heads exhibit higher alignment to the text representations than image representation averages.

vary the number of images $n$ per class used in each average ($n \in \{1, 5, 10, 20, 30, 40, 50\}$, with 50 representing the full validation set per class). Figure 5 shows that row vectors from the classification head matrix consistently achieve higher $m_{NN}$ scores than averaged image embeddings across all values of $n$ and neighborhood sizes $k \in \{3, 5, 10\}$. This suggests that classification heads exhibit stronger semantic alignment with text embeddings than image prototypes, supporting our approach of treating them as semantic representations for vision-language alignment.

**Modality Gap Between Classification Head Weights and Image Representations.** Inspired by the text-image modality gap observed in models like CLIP (Liang et al., 2022), we investigate whether a similar separation exists between classification weights $w_i$s and image representations. Following the analysis procedure from the modality gap seminal work Liang et al. (2022), our findings confirm a statistically significant gap: a permutation test on distances between centroids of both modalities showed that the observed distance (0.1723) was substantially larger than expected by chance (mean 0.0446, p<0.001). Furthermore, a simple single-layer MLP trained to distinguish between the two representation types achieved near-perfect test accuracy (99.75%). This systematic separation suggests that despite seman-

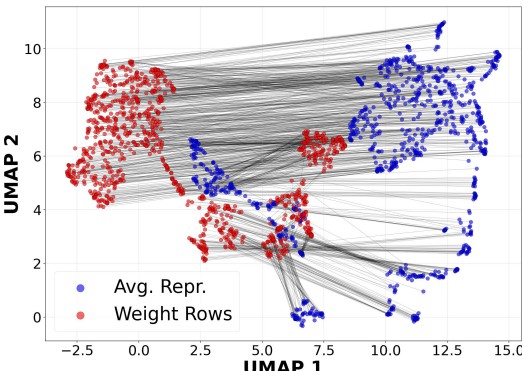

Figure 6: UMAP (McInnes et al., 2018) of class representations. Blue and red points show average image representation and weights $w_i$ respectively. Lines connect pairs from the same class.

tic alignment, classification weights and image features occupy distinct regions in the latent space. Figure 6 displays a UMAP (McInnes et al., 2018) visualization of weights and image representations where a separation between the two modalities can be observed. Extended details are provided in H.

## 5 DISCUSSION

Our approach enables the upgrade of a vision model into a vision-language model with minimal data and computational effort by recycling the usually discarded classification head weights from ImageNet-21K pretraining.

**Recycling for green AI.** Should we want specific image and text encoders to produce aligned embeddings, our approach avoids training from scratch a text encoder to be semantically aligned with the image encoder as in LiT (Zhai et al., 2022). We showed how minimal image-caption data (non-labelled) can improve the zero-shot classification accuracy in specific domains as satellite images, without hurting the model's accuracy in other tasks. By employing pretrained classification head weights as representations, we save on computation and manual labelling effort, thereby pushing progress towards the goals of green AI.

**Limitations and future work.** We aligned image and text encoders by combining image-text pairs and classification heads in the same dataset, as our goal was to demonstrate the effectiveness of this conceptually simple approach. However, there are caveats to treating image and weight representations on equal footing. For instance, we showed that an analogue to the well-known image-text modality gap (Liang et al., 2022) also occurs in the image representation space between images and row vectors $w_i$s of the classification head matrix. Although we presented preliminary attempts to mitigate this gap in Appendix H (e.g., via centering and rescaling or lightweight projections), these simple strategies did not result in performance gains. Thus, investigating this phenomenon further and finding more sophisticated approaches for combining image-text pairs and classification weight representations remains an interesting future research direction to further improve this post-hoc alignment augmentation approach.

In addition, our zero- and few-shot experiments employed a simple MLP to map text representations into the image representation space. Our intention was not to find the optimal post-hoc alignment method for these downstream tasks, but rather to demonstrate how augmenting image-text pairs with weight data can yield a competent vision-language model even with a simple alignment technique. We believe that more advanced alignment methods can benefit from augmenting alignment data with pretrained classification head weights, as demonstrated in our cross-modal retrieval experiments.

Lastly, beyond the applications demonstrated in this work, a vision-language model enables capabilities that vision-only backbones cannot provide. For example, one could explore prompt tuning methodologies such as CoOp (Zhou et al., 2022b) and CoCoOp (Zhou et al., 2022a)), Visual QA (Song et al., 2022) or image captioning with text-only training (Li et al., 2023). Our augmentation approach opens the door to apply these methods to vision-only models in a fast, efficient way.

## 6 CONCLUSION

We introduce a resource-efficient approach that recycles classification head weights from ImageNet-21K pretraining to align independently trained image and text encoders with minimal paired data. By leveraging these typically discarded weights as semantic prototypes, our approach improves post-hoc alignment techniques in cross-modal retrieval and demonstrates promising results in few- and zero-shot classification, even surpassing CLIP on CIFAR benchmarks without using any image-text representation pair for training. We provide empirical evidence for using classification heads as semantic prototypes by evaluating both the alignment to text representations and the accuracy of cosine similarity classifiers on classification-head representations. Future work could address the modality gap between weight and image representations, explore more sophisticated strategies to combine these modalities, and evaluate the approach in additional domains, alignment techniques, and downstream tasks. Overall, our approach reduces computational and data requirements for vision-text alignment, contributing towards more sustainable AI practices in multimodal learning.

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

## A  CODE

All code is available at: `https://anonymous.4open.science/r/Recycling4VLAlignment-0157/`

## B  DATASETS

**Retrieval.**  Evaluation of our data augmentation technique in retrieval setting is carried out on Flickr30K (Plummer et al., 2015), a widely-used benchmark for image-text retrieval tasks. The dataset contains 31,000 images collected from Flickr, each annotated with five human-written captions describing the visual content. To further validate the effectiveness of our approach, we also provide extended retrieval experiments on the MS-COCO dataset (Lin et al., 2014) in the appendix. MS-COCO is a large-scale dataset comprising over 120,000 images with diverse scenes and objects, each paired with five human-annotated captions. For both datasets, we follow the standard evaluation protocol using the widely-employed Karpathy splits (Karpathy & Fei-Fei, 2015).

**Classification.**  We evaluate our approach for zero- and few-shot classification tasks in nine diverse datasets spanning multiple domains including remote sensing, natural scenes, objects, animals, food, and textures. All datasets are used with their standard splits where available. We utilize the `torchvision` implementation for all datasets except RESISC45, which is obtained from the `torchgeo` library (Stewart et al., 2024).

**RESISC45** (Cheng et al., 2017) is a remote sensing image scene classification dataset containing 31,500 images across 45 scene classes, with 700 images per class at 256×256 pixel resolution.

**EuroSAT** (Helber et al., 2019) consists of 27,000 Sentinel-2 satellite images covering 10 land use and land cover classes across European cities, with each image having 13 spectral bands at 64×64 pixels.

**Flowers102** (Nilsback & Zisserman, 2008) contains 8,189 images of 102 flower categories commonly found in the United Kingdom, with high intra-class variation and inter-class similarity.

**OxfordPets** (Parkhi et al., 2012) includes 7,349 images of 37 cat and dog breeds, providing a fine-grained classification challenge for animal recognition.

**Food101** (Bossard et al., 2014) comprises 101,000 images across 101 food categories, with 1,000 images per class, representing popular dishes from around the world.

**CIFAR-10 and CIFAR-100** (Krizhevsky, 2009) are widely-used benchmark datasets containing 60,000 32×32 color images. CIFAR-10 has 10 classes while CIFAR-100 has 100 classes, both with 6,000 and 600 images per class respectively.

**DTD** (Cimpoi et al., 2014) (Describable Textures Dataset) contains 5,640 texture images organized into 47 categories, focusing on describable texture attributes rather than materials.

**Places365** (Zhou et al., 2018) is a scene-centric database designed for scene recognition and understanding. We use the Places365 validation set (50 images per class) instead of the larger test set (900 images per class) for computational efficiency. For zero-shot experiments, we evaluate on all 50 images per class. For few-shot experiments, we split each class into 10 training and 40 test images, then sample $N \in \{1, 2, 4, 8\}$ shots per class from the training set and evaluate on the 40 test images in each class.

## C  METRICS

In this work, we evaluated performance on both retrieval and classification tasks. Below we define each metric.

**Retrieval.**  For image–text and text–image retrieval, we employ the same standard metrics as in related work (Li et al., 2025).

Mean Average Precision (mAP) Mean Average Precision summarizes the area under the precision–recall curve across all queries. For a single query $q$,

$$\text{AP}(q) = \frac{1}{R_q} \sum_{k=1}^{N} P(k) \, \mathbf{1}\{\text{relevant at rank } k\},$$

where $R_q$ is the number of relevant items for $q$, $P(k)$ is precision at cutoff $k$, $N$ is the length of the retrieved list and $\mathbf{1}\{\cdot\}$ is the indicator function that equals 1 if the item at rank $k$ is relevant, and 0 otherwise. The mAP over $Q$ queries is then

$$\text{mAP} = \frac{1}{Q} \sum_{q=1}^{Q} \text{AP}(q).$$

Note: In the Flickr30k dataset each image has 5 associated captions, so $R_q = 5$ for each image query.

Precision at $K$ (P@K) Precision at rank $K$ measures the proportion of relevant items among the top-$K$ results:

$$P@K = \frac{\text{Number of relevant items in top } K}{K}.$$

We report:

- *Image→Text*: $P@1$ and $P@5$
- *Text→Image*: $P@1$

While we compute these retrieval metrics under both baseline and augmented conditions, in the main article we present metric gains (e.g. increase in mAP, and in P@1/5). This succinctly highlights the impact of our augmentation strategy.

**Classification accuracy** For our classification experiments, we report *Accuracy*, defined as

$$\text{Accuracy} = 100 \times \frac{\text{Number of correctly classified samples}}{\text{Total number of samples}}$$

Accuracy is the most intuitive and widely used metric for balanced multi-class settings.

# D  CROSS-MODAL RETRIEVAL

In the cross-modal retrieval experiment evaluate how three different alignment methods (CSA, text-to-concepts and MLP-alignment) improve their performance in the cross-modal retrieval downstream task when combined with our data augmentation approach. Here, we provide technical information and complement the results with additional aligned vision-language models. Specifically image encoders are upgraded to vision-language models to perform cross-modal retrieval using the CLIP text encoder.

First, we provide some further information on the hyperparameter choices for the different alignment methodologies. MLP-alignment consists of learning a lightweight MLP to map text to image representations. The lightweight MLP consists of a two-layer multilayer perceptron with GELU (Hendrycks & Gimpel, 2016) activation that maps text representations into the image feature space. The hidden dimension is 4 times that of the input dimension. The MLP is trained using the AdamW (Loshchilov & Hutter, 2019) optimizer with an initial learning rate of $5 \cdot 10^{-3}$ and cosine loss. We found that performance for retrieval and classification downstream tasks plateaued after around 500 epochs, with little change when training for longer, so we fixed the number of epochs to 500. We anneal the learning rate following a cosine schedule without restarts (Loshchilov & Hutter, 2017). Regarding the hyperparameters of CSA and text-to-concepts methods, the CSA method (Li et al., 2025) is implemented with the dimension of the shared representation space equal to 200 as we found out this was where downstream performance in Flickr30K cross-modal retrieval peaked. Similarly, the text-to-concept method is trained with MSE loss using an AdamW with cosine annealing schedule for 500 epochs. Since only lightweight MLPs are trained (two layers in the case of MLP-alignment and a single layer for text-to-concepts), the required training computational power is minimal. All experiments have been conducted on a GTX Titan Xp 12GB GPU where these lightweight models take around a couple of minutes to train.

Table 4: Cross-modal retrieval in Flickr30K for BEiT-B/16 aligned to CLIP-ViT-B/32's text encoder for different alignment methods (with and without weight augmentation) and varying training sample sizes. Metrics evaluated are: Image to text Mean Average Precision (mAP), Top-1 Precision (P@1) and Top-5 Precision (P@5) and Text to Image Top-1 Precision (P@1).

| Img. encoder | Metric | Alignment method | +ImgNet 21K | Training set size | | | | |
|---|---|---|---|---|---|---|---|---|
| | | | | 0 | 1K | 5K | 10K | 30K |
| **BEiT-B/16** | i2t mAP | CSA | No | | 0.0668 | 0.3991 | 0.4561 | 0.4985 |
| | | | Yes | **0.3860** | **0.4603** | **0.4829** | **0.4926** | **0.5067** |
| | | MLP | No | | 0.3702 | 0.5090 | 0.5481 | 0.5903 |
| | | | Yes | **0.4404** | **0.4945** | **0.5500** | **0.5730** | **0.6011** |
| | | Text2cpts | No | | 0.4316 | 0.5563 | 0.5758 | **0.5895** |
| | | | Yes | **0.4271** | **0.5243** | **0.5654** | **0.5782** | 0.5893 |
| | i2t P@1 | CSA | No | | 0.0690 | 0.5110 | 0.5780 | 0.6410 |
| | | | Yes | **0.4980** | **0.5900** | **0.6180** | **0.6500** | **0.6500** |
| | | MLP | No | | 0.4450 | 0.6260 | 0.6730 | 0.7160 |
| | | | Yes | **0.5370** | 0.6240 | **0.6720** | **0.6950** | **0.7300** |
| | | Text2cpts | No | | 0.5270 | **0.6910** | **0.7100** | 0.7370 |
| | | | Yes | **0.4990** | **0.6430** | 0.6870 | 0.7080 | **0.7380** |
| | i2t P@5 | CSA | No | | 0.0558 | 0.3560 | 0.4074 | 0.4514 |
| | | | Yes | **0.3390** | **0.4120** | **0.4290** | **0.4364** | **0.4520** |
| | | MLP | No | | 0.3246 | 0.4534 | 0.4920 | 0.5260 |
| | | | Yes | **0.3904** | **0.4430** | **0.4944** | **0.5124** | **0.5386** |
| | | Text2cpts | No | | 0.3856 | 0.4976 | 0.5180 | **0.5252** |
| | | | Yes | **0.3754** | **0.4690** | **0.5090** | **0.5206** | 0.5240 |
| | t2i P@1 | CSA | No | | 0.0038 | 0.1070 | 0.1428 | **0.2432** |
| | | | Yes | **0.1242** | **0.1488** | **0.1804** | **0.1850** | 0.2244 |
| | | MLP | No | | 0.2766 | 0.3548 | 0.3766 | **0.3964** |
| | | | Yes | **0.3350** | **0.3406** | **0.3742** | **0.3786** | 0.3926 |
| | | Text2cpts | No | | 0.2786 | 0.3486 | 0.3574 | 0.3622 |
| | | | Yes | **0.3160** | **0.3404** | **0.3580** | **0.3628** | **0.3636** |

For CSA, the heaviest operation is an SVD computation, which does not require GPU. All vision models are obtained from the checkpoint associated to ImageNet-21K task in `Timm` (Wightman, 2019) library. Results are shown in Table 4 (BEiT-B/16), Table 10 (CAFormer-S18), Table 11 (ConvFormer-S18), Table 12 (ConvNeXt-Base), and Table 13 (TinyViT-21M). The checkpoint for the text encoder of CLIP ViT-B/32 model is that from `torch.hub`.

**Zero-shot cross-modal retrieval**. We now focus on the zero-shot retrieval performance, where no task-specific training data is used, as shown in Table 3 for the BEiT-B/16 model aligned to the CLIP ViT-B/32 text encoder. The Flickr30K results correspond to those reported in Table 4 under the 0 training size column, and align with the x-axis equal to 0 in Figure 2. Ad-

Table 3: Zero-shot cross-modal retrieval for different alignment methods.

| Metric | Alignment Method | Datasets | |
|---|---|---|---|
| | | Flickr30K | COCO |
| i2t mAP | CSA | 0.3860 | 0.1842 |
| | MLP | 0.4404 | 0.2075 |
| | Text2cpts | 0.4271 | 0.2092 |
| i2t P@1 | CSA | 0.4980 | 0.2442 |
| | MLP | 0.5370 | 0.2562 |
| | Text2cpts | 0.4990 | 0.2540 |
| i2t P@5 | CSA | 0.3390 | 0.1668 |
| | MLP | 0.3904 | 0.1854 |
| | Text2cpts | 0.3754 | 0.1846 |
| t2i P@1 | CSA | 0.1242 | 0.0570 |
| | MLP | 0.3350 | 0.1556 |
| | Text2cpts | 0.3160 | 0.1433 |

ditionally, we provide results on the COCO dataset (Karpathy split), which we had not tested before, to further validate the generalization of our approach. Across both datasets and all three alignment methods, the models demonstrate non-trivial retrieval performance despite being trained exclusively on ImageNet-21K weight representations. These results validate that meaningful zero-shot cross-

modal retrieval capabilities can be achieved using only recycled weight representations, without requiring any image-text paired data.

**Weights vs image-text pairs for downstream retrieval**. As mentioned in the main text, a rightful concern is whether our model's improved retrieval performance is simply due to the addition of new data for alignment, rather than the intrinsic quality of the classification weights. To isolate this effect, we conducted an ablation study aligning the BEiT-B/16 image-encoder to text using MLP-alignment (see Section 4). We compared alignment using 1K class weight representations against alignment using the same number of ImageNet image-text pairs. Both models are aligned using only this 1K data and then evaluated on the downstream Flickr30k retrieval task in a zero-shot setting. Results presented in Table 5, reveal that the model aligned using recycled classification weights substantially outperforms the one aligned with image-text pairs. This result provides evidence that the classification weights provide a richer, higher-quality signal for alignment than image-text pairs, which is reflected even in downstream image-text retrieval.

| Metric | Weight repr. | Image repr. |
|--------|-------------|-------------|
| i2t mAP | **0.2751** | 0.1860 |
| i2t P@1 | **0.3230** | 0.2110 |
| i2t P@5 | **0.2388** | 0.1566 |
| t2i P@1 | **0.2212** | 0.1406 |

Table 5: Flickr30k zero-shot retrieval performance. We compare MLP-alignment recycling 1K ImageNet weight representations against the same number of image-text pairs representations.

## E  ZERO-SHOT CLASSIFICATION

This experiment expands the zero-shot classification evaluation from the Experiments section in the paper by assessing aligned models zero-shot performance across a broader range of text encoders. We use MLP-alignment introduced in the retrieval experiment, which consists on learning a lightweight MLP to map text to image representations. The architecture, hyperparameters and training procedure of this lightweight MLP are exactly the same as those explained in the cross-modal retrieval (Appendix D). During inference, zero-shot classification is done by computing:

$$\arg \max_{1 \leq i \leq C} \big\{ \cos\big(f_I(x),$$
$$\text{MLP} \circ f_T(\textit{A photo of} <class>)\big)\big\} \tag{7}$$

where $f_I(x)$ represents the image encoder features for image $x$ and $\text{MLP} \circ f_T$ denotes the aligned text encoder applied to template-based class descriptions.

We conduct experiments across the nine classification benchmarks introduced in Appendix B. Five image encoders are evaluated (BEiT-B/16 (Bao et al., 2021), CAFormer-S18, ConvFormer-S18, ConvNeXt-B, and TinyViT-21M) when paired with four text encoders, CLIP ViT-B/32 text encoder, RoBERTa (all-roberta-large-v1), MPNet (all-mpnet-base-v2), and MiniLM (all-MiniLM-L6-v2), with the latter three from the `sentence-transformers` library. We also evaluated ResNet and ViT CLIP vision-language models (Radford et al., 2021), which were trained on >400M image-text pairs to achieve aligned representations.

Results presented in Table 7 show zero-shot accuracy when only ImageNet-21K pretraining classification weights are used to train the aligner MLP. The findings reveal consistent patterns across image encoders: the CLIP ViT-B/32 text encoder substantially outperforms text-only trained models, achieving best results in almost all configurations with performance gaps often exceeding 20-40 percentage points. Within each image encoder family, MPNet typically ranks second, followed by RoBERTa, while MiniLM generally shows the weakest performance. This superiority of accuracy could indicate CLIP's text encoder has learned to represent language in a way that's inherently connected to visual concepts through its training on image-caption pairs. The visual semantic understanding remains transferable even when the CLIP's text encoder is paired with completely different image encoders from the one it was originally trained with. This suggests that CLIP's advantage may not stem solely from jointly optimizing vision and text encoders, but specifically from CLIP's text encoder having learned to represent visual concepts in a much more effective way. In contrast, sentence-transformer models, despite excelling at capturing semantic relationships in text, lack this visual grounding, since they were trained purely on textual data without any visual context.

Table 6, which presents the data from Figure 3, and Table 7 can be used to we compare on zero-shot classification accuracy CLIP's natively aligned image and text encoders, which were trained on approximately 400 million image-caption pairs to produce joint representations, to encoders aligned without using image-caption pairs but the 21K weight vectors from ImageNet-21K pretraining. Despite the substantial difference in training data to align image and text modalities—orders of magnitude fewer paired examples—MLP-aligned image and text encoders, achieve competitive performance on several zero-shot classification tasks and notably strong results on CIFAR10 and CIFAR100, where most aligned classifiers surpass the CLIP variants. The lowest performance for MLP-aligned image and text encoders in zero-shot classification is observed on datasets such as RE-SISC45 and EuroSAT, likely due to the minimal overlap between satellite or remote-sensing imagery and the ImageNet distribution.

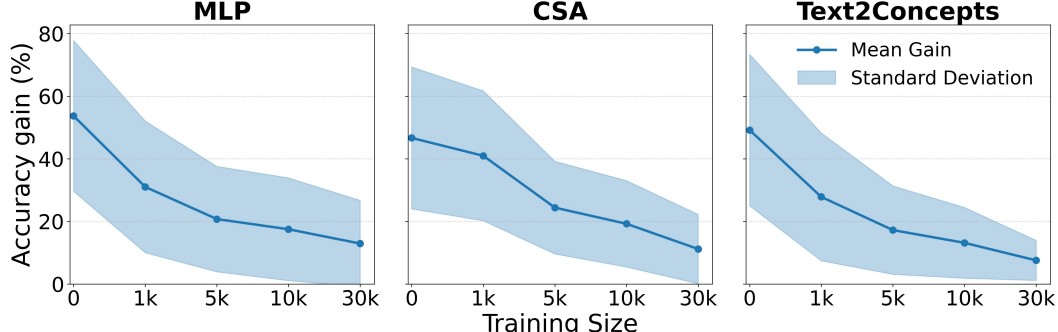

Figure 7: Mean zero-shot accuracy gains for BEiT-B/16 image encoder aligned to text on Flickr30k image-text pairs vs when augmenting alignment data with recycled ImageNet21K weights representation. Zero-shot accuracy evaluated on: RESISC45, EuroSAT, Flowers102, OxfordPets, Food101, CIFAR-10, CIFAR-100, DTD, and Places365. The shaded region indicates the standard deviation across datasets.

Table 6: Zero-shot classification accuracy. BEiT-B/16 encoders are aligned to text via MLP training using different representations: ImageNet1K classification head weights, ImageNet21K's, and ImageNet21K's enhanced with one image-caption pair per class. CLIP ViT-B/16 results are included as a reference, representing a natively text-aligned encoder trained on WIT400M (≈400M image-text pairs) with extensive computational resources (Radford et al., 2021). Bold indicates best performance per dataset; underlined indicates second-best.

| Image Encoder | Alignment train data | RESISC | EUROSAT | FLOWERS | PETS | FOOD | CFR10 | CFR100 | DTD | PLACES |
|---|---|---|---|---|---|---|---|---|---|---|
| BEiT-B/16 | IN1K repr. | 21.16 | 17.70 | 11.82 | 75.69 | 20.96 | 89.35 | 61.12 | 22.71 | 24.32 |
| | IN21K repr. | 24.21 | 23.67 | 62.63 | 78.06 | 57.12 | **94.40** | 73.66 | 35.37 | 34.34 |
| | IN21K repr. + 1 img-capt. pair per class | 32.37 | 30.85 | **73.74** | 80.98 | 64.76 | 94.11 | **74.04** | 38.94 | 34.78 |
| CLIP ViT-B/16 | WIT400M | **57.14** | **51.93** | 65.96 | **88.31** | **86.04** | 89.67 | 68.05 | **43.99** | **38.96** |

**Zero-shot classification accuracy in a specialized domain dataset**. We evaluate the zero-shot performance of our post-hoc alignment on a specialized domain exhibiting a significant distribution shift from ImageNet-21K. We employ the HAM10000 dataset (Tschandl et al., 2018), a benchmark for dermatological skin lesion classification comprising 7 classes. Table 9 compares our method (BEiT-B16 aligned via recycled ImageNet-21K weights) against ViT CLIP baselines and a random classification baseline. We report Balanced Accuracy to account for class imbalance. The results indicate that the specialized nature of dermatological images presents a challenge for all evaluated Vision-Language Models. Both CLIP and our proposed method achieve performance only marginally above the random baseline (14.28%). In particular, our method achieves a balanced accuracy of 18.32%, similar to the CLIP ViT-B/32 variant (18.85%). This suggests that the alignment learned from ImageNet-21K weights is as robust as CLIP's text-image alignment when applied to unseen specialized domains. In addition, it confirms that for such specific tasks, fine-tuning is necessary for any generalist VLM, whether trained end-to-end like CLIP or post-hoc aligned.

**Visualizing the Impact of Weight Recycling on Alignment**. To visually investigate how incorporating ImageNet-21K weights affects the geometric configuration of embeddings from an MLP-aligned model, we visualize the latent space geometry using UMAP McInnes et al. (2018). Figure 10 displays the embeddings of CIFAR-100 class texts (MLP($f_T$('A photo of $< class\ i >$'))) and the average image representations of the corresponding classes (via the image encoder $f_I$). In the random alignment baseline (Figure 10a), the representations present no structure; specifically, a text representation of a class is not closer to its corresponding image representation than to any other image representation. When training the alignment MLP using ImageNet-1K weights (Figure 10b), we observe an organization in the spatial distribution. We note that this configuration results in a downstream accuracy in zero-shot classification in CIFAR-100 of less than 60% (see Figure 3). However, when training with ImageNet-21K weights (Figure 10c), we observe a significantly more structured configuration. In this case, the downstream accuracy rises to 73.66% (see Table 7 and Figure 3), demonstrating that the richer semantic information in the 21K weights leads to better alignment in this downstream task.

Table 7: Zero-shot classification accuracy (%) of image and text encoders aligned with an MLP that was trained only on classification heads of ImageNet-21K pretraining. Checkpoint of text models: CLIP = Text encoder of CLIP ViT-B/32 model in `torch.hub`, RoBERTa = all-roberta-large-v1, MPNet = all-mpnet-base-v2, MiniLM = all-MiniLM-L6-v2. Best result for each image encoder in each dataset is shown in **bold** and second-best is underlined. For reference, we evaluated ResNet and ViT CLIP vision-language models (Radford et al., 2021), which were trained on ≈400M image-text pairs to achieve aligned representations.

| Img. encoder | Txt. encoder | RESISC | EUROSAT | FLOWERS | PETS | FOOD | CFR10 | CFR100 | DTD | PLACES |
|---|---|---|---|---|---|---|---|---|---|---|
| BEiT-B/16 | CLIP | **24.21** | **23.67** | **62.63** | **78.06** | **57.12** | **94.40** | **73.66** | **35.37** | **34.34** |
| BEiT-B/16 | RoBERTa | 18.54 | 17.15 | 38.54 | 35.32 | 29.48 | 92.12 | 65.93 | 16.28 | 25.86 |
| BEiT-B/16 | MPNet | 20.95 | 18.15 | 35.75 | 36.14 | 27.20 | 92.70 | 67.37 | 18.78 | 26.29 |
| BEiT-B/16 | MiniLM | 14.35 | 12.85 | 31.86 | 28.48 | 25.24 | 92.29 | 63.58 | 15.85 | 20.61 |
| CAFormer-S18 | CLIP | 16.89 | **22.70** | **67.44** | **75.20** | **59.83** | 86.05 | **69.08** | **37.39** | **32.85** |
| CAFormer-S18 | RoBERTa | 16.11 | 20.70 | 40.80 | 33.96 | 34.54 | **89.33** | 66.15 | 12.87 | 27.33 |
| CAFormer-S18 | MPNet | **18.52** | 20.48 | 42.92 | 46.96 | 32.69 | 86.76 | 67.36 | 15.05 | 25.28 |
| CAFormer-S18 | MiniLM | 11.27 | 17.37 | 38.87 | 37.97 | 20.10 | 85.64 | 62.09 | 15.21 | 20.27 |
| ConvFormer-S18 | CLIP | **19.71** | 23.93 | **66.12** | **75.47** | **54.92** | 85.03 | **65.85** | **37.82** | **32.67** |
| ConvFormer-S18 | RoBERTa | 17.33 | 16.44 | 35.47 | 38.81 | 32.30 | **85.98** | 62.40 | 12.71 | 26.57 |
| ConvFormer-S18 | MPNet | 18.52 | **27.00** | 33.62 | 45.68 | 30.61 | 82.31 | 59.09 | 16.65 | 24.63 |
| ConvFormer-S18 | MiniLM | 12.73 | 18.59 | 30.25 | 38.18 | 17.81 | 83.05 | 57.32 | 12.87 | 19.29 |
| ConvNext-B | CLIP | **27.29** | **24.78** | **69.21** | 73.78 | **60.29** | **92.34** | **71.66** | **35.85** | **35.09** |
| ConvNext-B | RoBERTa | 22.79 | 19.15 | 29.81 | 25.48 | 28.75 | 90.99 | 67.04 | 16.91 | 27.96 |
| ConvNext-B | MPNet | 24.65 | 23.63 | 40.80 | 42.00 | 30.52 | 91.81 | 68.68 | 19.73 | 28.48 |
| ConvNext-B | MiniLM | 21.73 | 17.00 | 39.70 | 26.27 | 27.55 | 90.97 | 64.64 | 18.83 | 22.61 |
| TinyViT-21M | CLIP | **28.33** | **26.48** | **68.42** | **76.81** | **57.30** | **89.69** | **69.39** | **35.59** | **33.80** |
| TinyViT-21M | RoBERTa | 22.75 | 21.48 | 34.90 | 26.76 | 22.09 | 86.94 | 65.05 | 14.84 | 25.91 |
| TinyViT-21M | MPNet | 24.59 | 25.56 | 33.22 | 25.59 | 26.59 | 88.98 | 64.22 | 18.94 | 26.09 |
| TinyViT-21M | MiniLM | 18.68 | 24.93 | 32.41 | 27.56 | 20.52 | 87.46 | 59.94 | 15.37 | 20.89 |
| CLIP RN-101 | | 41.37 | 26.70 | 61.99 | 86.24 | 79.83 | 77.77 | 48.52 | 39.68 | 36.00 |
| CLIP RN-50 | | 41.25 | 28.00 | 61.12 | 85.15 | 75.99 | 69.81 | 40.68 | 39.73 | 36.96 |
| CLIP ViT-B/16 | | 57.14 | 51.93 | 65.96 | 88.31 | 86.04 | 89.67 | 68.05 | 43.99 | 38.96 |
| CLIP ViT-B/32 | | 55.19 | 34.44 | 63.64 | 87.76 | 80.53 | 89.08 | 64.14 | 29.79 | 38.71 |

# F  FEW-SHOT CLASSIFICATION

We evaluate our few-shot classification approach on the nine diverse classification tasks across standard shot settings with 1, 2, and 4 shots per class, comparing against Nearest Centroid Classifier (NCC) and K-Nearest Neighbors (KNN) baselines on frozen backbones. NCC has been shown to be remarkably effective for few-shot classification, often surpassing more complex meta-learning approaches (Luo et al., 2023). Our sequential training methodology employs the same lightweight MLP architecture as used in cross-modal retrieval and zero-shot classification settings, trained with identical optimizer, scheduler, and loss configurations from previous experiments. The initial alignment stage trains for 500 epochs using only ImageNet-21K classification weights, followed by 200 epochs of fine-tuning on target dataset image-text pairs. Once the MLP is trained, we predict the class of a given image $x$ with Equation (7). To assess statistical significance of accuracy improvements, we conduct five independent experimental runs where the exact same images are sampled

for each class across all compared methods (ours, NCC, KNN) within each run, ensuring controlled comparison. We then perform paired t-tests on the resulting accuracy differences between methods, assuming normality of the accuracy distribution across runs, to rigorously validate the statistical significance of our method's gains over NCC and KNN. As noted in the main article, the classification method based on sequential training and Equation (7) is probably suboptimal. However, it serves to illustrate that even a naive approach can be competitive when combined with our weight augmentation method.

# G   CLASSIFICATION HEAD WEIGHTS ROW VECTORS AS PROTOTYPES FOR THE COSINE CLASSIFIER

In the main text, we provided empirical evidence (see Table 2) that the row vectors $w_i$ of a classification head $W$ can serve as effective class prototypes for a cosine classifier. We also briefly mentioned the Neural Collapse (NC) phenomenon (Papyan et al., 2020) as a potential theoretical basis, although the gaps between the ideal assumptions in NC and our setting lead us to provide experimental results as evidence. This appendix provides a more detailed discussion for using the classification weights $w_i$ as class prototypes in the cosine classifier, proceeding in two parts: first, an intuitive view, and second, a more formal justification via the properties of Neural Collapse.

**An intuitive view with degrees of freedom**.   A linear classifier on a point $x$ is defined by $\text{argmax}_i w_i^T x + b_i$. The decision is based on the *direction* of the weight vector $w_i$, its *magnitude* (norm) $||w_i||$, and the *bias* scalar $b_i$. On the other hand, in the cosine classifier the decision rule becomes

$$\text{argmax}_i \cos(w_i, x) = \text{argmax}_i \left( \frac{w_i^T x}{||w_i||||x||} \right) = \text{argmax}_i \left( \frac{w_i^T x}{||w_i||} \right) \tag{8}$$

In Equation (8), the norm and the bias take no part in the classification, as only the *directional* information of $w_i$ matters. For a $d$-dimensional feature space (e.g., $d = 768$), the score for each class depends on $d + 1$ parameters ($w_i \in \mathbb{R}^d$ and $b_i \in \mathbb{R}$). The normalized vector $w_i/||w_i||$ retains $d - 1$ degrees of freedom (the direction on the hypersphere), "losing" only 2 parameters (the magnitude and bias). The central hypothesis, supported by our empirical results in Table 2, is that the vast majority of the semantic, class-identifying information is encoded in the *direction* of this $d$-dimensional vector, not its magnitude or bias. The theoretical framework of Neural Collapse formalizes this intuition.

**Formal explanation via Neural Collapse (NC)**. The Neural Collapse (NC) phenomenon (Papyan et al., 2020) describes the geometric structure of the final-layer features and classifier weights during the terminal phase of cross-entropy training (i.e., after achieving near-zero training error). The key properties for us are[1]:

- *(NC1) Variability Collapse:* The feature vectors $x$ for all training samples belonging to a class $i$ collapse to their class mean, or centroid, $\mu_i$.

- *(NC3) Convergence to Self-Duality:* The classifier's weight vectors $w_i$ (from the $Wx + b$ layer) converge to be their corresponding class centroids $\mu_i$ up to rescaling by a scalar.

- *(NC4) Simplification to Nearest Class-Center:* The $Wx + b$ classifier converges to be equivalent to a nearest-centroid classifier, i.e., $\text{argmax}_i(w_i^T x + b_i) \rightarrow \text{argmin}_i ||x - \mu_i||$

We show these conditions lead to an equivalence between the linear classifier and the cosine classifier using $w_i$s rows as representations. First, we assume as in our setting, our features $x$ to be L2-normalized, so $||x|| = 1$. From *(NC1)*, all features in a class $i$ collapse to their centroid $\mu_i$, which implies in the training limit the centroids themselves must also be normalized (as they are equal to any of the representations of the points in the class), so $||\mu_i|| = 1$. From *(NC3)*, the weights $w_i$ align with these centroids $\mu_i$; given that $||\mu_i|| = 1$, the "up to rescaling" property simplifies, and the *normalized weight vector* and the class *centroid* converge to each other: $\mu_i \rightarrow w_i/||w_i||$. Finally, from *(NC4)*, the $Wx + b$ classifier converges to a nearest-centroid classifier, $\text{argmax}_i(w_i^T x + b_i) \rightarrow \text{argmin}_i ||x - \mu_i||^2$. By substituting the result from *(NC3)* into *(NC4)*, we see that the $Wx + b$

---

[1]NC2 refers to the geometric configuration of class centroids and classifier's weight vectors, which is not relevant for our purposes.

classifier, at convergence, becomes equivalent to a classifier that finds the nearest *normalized weight vector*: $\arg\max_i(w_i^T x + b_i) \to \arg\min_i ||x - w_i/||w_i|| \,||$. Now we see this is the same as the cosine classifier on the rows of the weight matrix. Since we assumed $||x|| = 1$, we have:

$$\left\| x - \frac{w_i}{||w_i||} \right\|^2 = ||x||^2 - 2\frac{w_i^T x}{||w_i||} + \left\| \frac{w_i}{||w_i||} \right\|^2 = 1 - 2\frac{w_i^T x}{||w_i||} + 1 = 2(1 - \frac{w_i^T x}{||w_i||}) \quad (9)$$

Hence

$$\arg\min_i \left\| x - \frac{w_i}{||w_i||} \right\| = \arg\min_i \left\| x - \frac{w_i}{||w_i||} \right\|^2 \overset{(9)}{=} \arg\min_i \left( 2(1 - \frac{w_i^T x}{||w_i||}) \right)$$
$$= \arg\max_i \left( \frac{w_i^T x}{||w_i||} \right) \overset{(8)}{=} \arg\max_i \cos{(w_i, x)} \quad (10)$$

Thus $\arg\max_i(w_i^T x + b_i) \to \arg\max_i \cos{(w_i, x)}$, that is, the linear classifier becomes the cosine classifier on the rows of the weight matrix. As noted in the main text, the formal theory of Neural Collapse relies on strong assumptions (e.g., balanced data, number of classes less than the feature dimension, zero training error). While recent extensions address some of these limitations individually by adding new technical conditions (Jiang et al., 2024; Yang et al., 2022), to the best of our knowledge, there are still gaps between theoretical assumptions and our setting. This is why the direct empirical validation in Table 2 remains a crucial piece of evidence supporting our approach.

## H    MODALITY GAP BETWEEN CLASSIFICATION HEAD WEIGHTS AND IMAGE REPRESENTATIONS

While vision-language models like CLIP demonstrate effective alignment between text and image representations in shared latent spaces, prior work has identified a modality gap where representations from image and text data occupy distinct regions (Liang et al., 2022). We investigate whether a similar phenomenon exists between row vectors $w^i$ of the classification head weight matrix $W$ and representations of images belonging to the corresponding classes. Specifically, we examine ImageNet-21K weights by extracting weight vectors for the 1K classes that correspond to ImageNet-1K, alongside averaged image representations computed from 50 images from each class in the ImageNet-1K. Following (Liang et al., 2022), we carry out two analyses: a permutation test of the centroid distances and an analysis of linear separability by training a single-layer MLP to distinguish between weight vectors and image representations. The permutation test evaluates whether the observed distance between modality centroids exceeds what would be expected by chance, by comparing the actual distance to a null distribution generated from random permutations of the data. Through this permutation testing of centroid distances, we observe a statistically significant separation: the distance between image and weight centroids (0.1723) substantially exceeds chance expectation (mean=0.0446, std=0.0014, p<0.001, Cohen's d=93.91). Additionally, by training a single-layer MLP classifier to discriminate between weight and image representations using an 80/20 train-test split, the classifier correctly classifies 199/200 test weight samples and 200/200 test image samples. These findings suggest that despite semantic alignment, there exists a systematic geometric separation between classification head row vectors and image features. As shown in Figure 8, the distribution of cosine similarities between weights and image representations (W-Img) is clearly shifted towards lower values compared to the distributions within each modality (W-W and Img-Img).

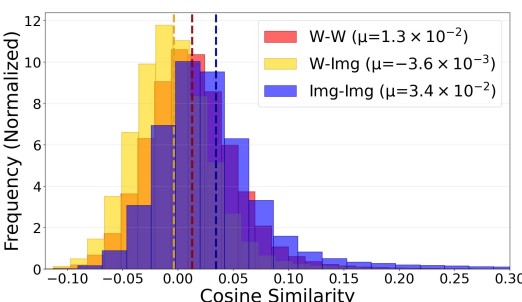

Figure 8: Distribution of the cosine similarities for averaged image representations and classification weights corresponding to the ImageNet1K classes. W-W indicates the distribution of cosine similarities between weights, Img-Img between average image representations, and W-Img inter-modality cosine similarities.

**Testing modality gap mitigation strategies**. Following the observation of the geometric separation between weight and image representations, we conducted preliminary experiments to determine if bridging this *modality gap* prior to alignment enhances downstream performance. We compared the baseline strategy (union from Equation (3)) against two explicit mitigation approaches applied to the ImageNet-1K weight representations:

- Centering and rescaling: we subtract the mean of the weight representations (centering), adding the mean of the image representations (shifting towards the image modality) and rescaling the vectors to unit norm.
- Lightweight projection: we introduced lightweight linear projection layers designed to map weight representations onto the image manifold, trained alongside the alignment objective.

Figure 9 describes the increase in downstream zero-shot classification and retrieval performance across multiple benchmarks when modality-gap mitigation strategies are employed in alignment data. Our results indicate that the modality gap mitigation strategies do not improve downstream performance compared to the union approach from Equation (3). This is not surprising, as it aligns with observations in (Liang et al., 2022), which note that closing the modality gap does not necessarily guaranty better downstream performance. Consequently, since these mitigation techniques did not yield performance gains, we maintain the use of the original union strategy for this work and leave the exploration of more advanced gap mitigation techniques for future research.

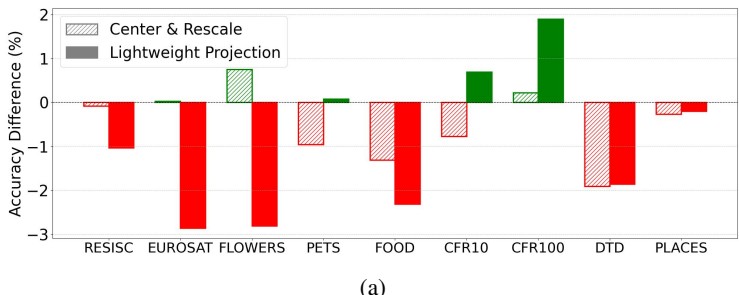

(a)

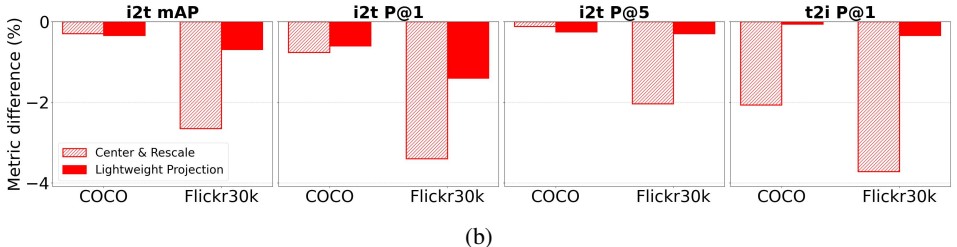

(b)

Figure 9: Change in zero-shot classification (a) and retrieval (b) performance when using basic modality gap mitigation strategies. ImageNet-1K weights representations undergoing two different modality-gap mitigation strategies (*Center & Rescale* and *Lightweight projection*) are used as alignment data. The results show that explicitly mitigating the geometric modality gap did not result in better downstream performance.

## I   ALIGNMENT USING CLASSIFIER WEIGHTS FROM A DOMAIN-SPECIFIC DATASET

In this section, we extend the evaluation of our proposed method by exploring how zero-shot classification accuracy varies when utilizing as alignment data weight representations from pretraining in domain-specific dataset iNaturalist (Van Horn et al., 2018). iNaturalist is a large-scale dataset characterized by fine-grained categories of flora and fauna, presenting a highly specialized semantic domain. The ConvNeXt checkpoint used as the image encoder was obtained from the `Timm` (Wightman, 2019) library.

As shown in Table 8, the downstream performance is notably higher when using ImageNet21K. This superior performance is likely due to the greater semantic overlap between the diverse ImageNet21K classes and the classes found in the downstream datasets. In contrast, the semantic variance within iNaturalist is limited to the biomedical domain. Consequently, it is reasonable to hypothesize that the alignment of the text and image encoders in the ImageNet21K case is performed over a much broader and diverse region of the latent space. This hypothesis is supported by the observation that the iNaturalist-aligned models exhibit a clear domain preference: they achieve relatively higher accuracy on datasets containing plant and animal classes (such as Flowers102, Oxfordpets, and CIFAR) (Nilsback & Zisserman, 2008; Parkhi et al., 2012; Krizhevsky, 2009) compared to non-biological datasets. However, it is important to note that despite this domain affinity, the absolute performance on these biological datasets remains lower than that of the ImageNet21K-aligned models. Finally, we note that performance on some of the datasets lacking specific plant or animal content remains non-trivial, suggesting that despite the imperfect alignment, a meaningful amount of general semantic information is still being transferred.

Table 8: A comparison of zero-shot classification accuracy (%) for iNaturalist and ImageNet21K weights serving as alignment data for training. We use ConvNeXt as image encoder.

| Alignment method | Data | RESISC | EUROSAT | FLOWERS | PETS | FOOD | CIFAR10 | CIFAR100 | DTD | PLACES |
|---|---|---|---|---|---|---|---|---|---|---|
| CSA | ImageNet21k | 20.60 | 18.93 | 52.20 | 60.70 | 50.05 | 90.56 | 65.46 | 27.71 | 30.72 |
| | iNaturalist | 4.68 | 16.33 | 37.62 | 20.85 | 5.28 | 30.07 | 14.46 | 6.76 | 1.11 |
| MLP | ImageNet21k | 27.29 | 24.78 | 69.21 | 73.78 | 60.29 | 92.34 | 71.66 | 35.85 | 35.09 |
| | iNaturalist | 2.48 | 15.81 | 24.26 | 8.42 | 2.19 | 34.34 | 6.80 | 2.87 | 0.43 |
| Text2cpts | ImageNet21k | 24.33 | 20.52 | 55.41 | 71.16 | 51.26 | 91.75 | 65.90 | 32.66 | 31.66 |
| | iNaturalist | 2.60 | 9.26 | 42.67 | 17.20 | 2.90 | 37.45 | 17.38 | 8.83 | 0.44 |

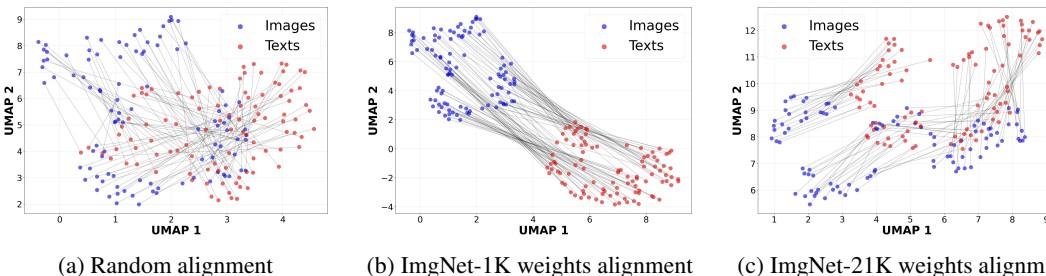

(a) Random alignment    (b) ImgNet-1K weights alignment    (c) ImgNet-21K weights alignm.

Figure 10: UMAP visualizations of text and average image embeddings in the CIFAR-100 dataset when using as aligner functions: (a) random projection function, (b) an MLP trained on ImageNet-1K weight representations, and (c) an MLP trained on ImageNet21K weight representations. We observe that the geometric configuration in (c) exhibits a more defined and complex structure compared to (b), which in turn presents more organization than (a).

Table 9: Zero-shot Balanced Accuracy on the HAM10000 Tschandl et al. (2018) dataset. We compare post-hoc alignment on BEiT-B16 with CLIP variants. We observe that all models struggle with this specialized domain, yet our weight-recycling approach performs on par with the CLIP variants.

| Model | Balanced Accuracy (%) |
|---|---|
| Random | 14.28 |
| CLIP ViT-B/32 | **18.85** |
| CLIP ViT-L/14@336px | 17.72 |
| BEiT-B16 (Ours) | 18.32 |

## J    LARGE LANGUAGE MODEL USAGE

Large Language Models (LLMs) were used in the preparation of this manuscript for writing assistance purposes. Specifically:

**Models Used:** OpenAI's ChatGPT and Anthropic's Claude were employed during the writing process.

**Role and Scope:** The LLMs were used exclusively for writing polishing and editorial assistance, including

- Grammar checking and correction
- Sentence restructuring for improved clarity and readability
- Style consistency improvements
- Word choice optimization

**Limitations:** The LLMs did not contribute to:

- Research ideation or conceptualization
- Experimental design or methodology
- Data analysis or interpretation of results
- Generation of novel technical content or insights
- Literature review or citation selection

All substantive intellectual contributions, including research ideas, experimental work, analysis, and conclusions, were developed independently by the authors. The final content, accuracy, and scholarly integrity of this work remain the sole responsibility of the listed authors.

Table 10: Cross-modal retrieval in Flickr30K for CAFormer-S18 aligned to CLIP-ViT-B/32's text encoder for different alignment methods (with and without weight augmentation) and varying training sample sizes. Metrics evaluated are: Image to text Mean Average Precision (mAP), Top-1 Precision (P@1) and Top-5 Precision (P@5) and Text to Image Top-1 Precision (P@1).

| Img. encoder | Metric | Alignment method | +ImgNet 21K | 0 | 1K | 5K | 10K | 30K |
|---|---|---|---|---|---|---|---|---|
| CAFormer-S18 | i2t mAP | CSA | No | | 0.0033 | 0.2936 | 0.3941 | 0.4582 |
| | | | Yes | **0.3182** | **0.4063** | **0.4275** | **0.4395** | **0.4635** |
| | | MLP | No | | 0.3197 | **0.4471** | 0.4950 | **0.5391** |
| | | | Yes | **0.3745** | **0.3833** | 0.4352 | **0.4963** | 0.5362 |
| | | Text2cpts | No | | 0.3974 | **0.5184** | 0.5360 | **0.5485** |
| | | | Yes | **0.3696** | **0.4727** | 0.5176 | 0.5311 | 0.5436 |
| | i2t P@1 | CSA | No | | 0.0020 | 0.3390 | 0.4970 | 0.5690 |
| | | | Yes | **0.4160** | **0.5020** | **0.5390** | **0.5440** | **0.5750** |
| | | MLP | No | | 0.3800 | **0.5460** | **0.6210** | **0.6730** |
| | | | Yes | **0.4440** | 0.4650 | 0.5260 | 0.6170 | 0.6600 |
| | | Text2cpts | No | | 0.4980 | 0.6400 | 0.6600 | **0.6780** |
| | | | Yes | **0.4700** | **0.6120** | **0.6470** | **0.6680** | 0.6700 |
| | i2t P@5 | CSA | No | | 0.0012 | 0.2640 | 0.3524 | 0.4120 |
| | | | Yes | **0.2818** | **0.3668** | **0.3816** | **0.3932** | **0.4168** |
| | | MLP | No | | 0.2770 | **0.3964** | 0.4404 | **0.4868** |
| | | | Yes | **0.3336** | **0.3400** | 0.3870 | **0.4418** | 0.4842 |
| | | Text2cpts | No | | 0.3496 | 0.4626 | **0.4796** | **0.4920** |
| | | | Yes | **0.3316** | **0.4268** | **0.4628** | 0.4754 | 0.4898 |
| | t2i P@1 | CSA | No | | 0.0018 | 0.0578 | 0.1616 | 0.2186 |
| | | | Yes | **0.0872** | **0.1260** | **0.1892** | 0.1772 | **0.2264** |
| | | MLP | No | | 0.2456 | 0.3474 | 0.3842 | 0.4124 |
| | | | Yes | **0.2892** | **0.2904** | **0.3752** | **0.3882** | **0.4126** |
| | | Text2cpts | No | | 0.2464 | **0.2946** | **0.3056** | **0.3140** |
| | | | Yes | **0.2160** | **0.2698** | 0.2892 | 0.2974 | 0.3098 |

Table 11: Cross-modal retrieval in Flickr30K for ConvFormer-S18 aligned to CLIP-ViT-B/32's text encoder for different alignment methods (with and without weight augmentation) and varying training sample sizes. Metrics evaluated are: Image to text Mean Average Precision (mAP), Top-1 Precision (P@1) and Top-5 Precision (P@5) and Text to Image Top-1 Precision (P@1).

| Img. encoder | Metric | Alignment method | +ImgNet 21K | Training set size | | | | |
|---|---|---|---|---|---|---|---|---|
| | | | | 0 | 1K | 5K | 10K | 30K |
| ConvFormer-S18 | i2t mAP | CSA | No | | 0.0027 | 0.2117 | 0.2933 | 0.3856 |
| | | | Yes | **0.3156** | **0.3979** | **0.4222** | **0.4396** | **0.4578** |
| | | MLP | No | | 0.3141 | **0.4468** | **0.4848** | **0.5321** |
| | | | Yes | **0.3956** | **0.3747** | 0.4392 | 0.4762 | 0.5302 |
| | | Text2cpts | No | | 0.3972 | 0.5145 | **0.5341** | **0.5475** |
| | | | Yes | **0.3807** | **0.4736** | **0.5174** | 0.5314 | 0.5436 |
| | i2t P@1 | CSA | No | | 0.0020 | 0.2510 | 0.3530 | 0.4840 |
| | | | Yes | **0.4130** | **0.5050** | **0.5520** | **0.5650** | **0.5680** |
| | | MLP | No | | 0.3820 | **0.5500** | 0.5840 | **0.6530** |
| | | | Yes | **0.4840** | **0.4660** | 0.5300 | **0.5940** | 0.6440 |
| | | Text2cpts | No | | 0.4940 | **0.6290** | 0.6520 | **0.6580** |
| | | | Yes | **0.4970** | **0.5940** | 0.6220 | 0.6470 | 0.6570 |
| | i2t P@5 | CSA | No | | 0.0006 | 0.1934 | 0.2646 | 0.3450 |
| | | | Yes | **0.2790** | **0.3544** | **0.3718** | **0.3912** | **0.4090** |
| | | MLP | No | | 0.2746 | **0.3928** | **0.4286** | **0.4796** |
| | | | Yes | **0.3498** | **0.3306** | 0.3912 | 0.4214 | 0.4786 |
| | | Text2cpts | No | | 0.3498 | 0.4582 | 0.4730 | **0.4894** |
| | | | Yes | **0.3396** | **0.4276** | **0.4670** | **0.4744** | 0.4884 |
| | t2i P@1 | CSA | No | | 0.0010 | 0.0652 | 0.0666 | 0.1424 |
| | | | Yes | **0.0976** | **0.1608** | **0.1642** | **0.2026** | **0.2144** |
| | | MLP | No | | 0.2432 | 0.3550 | 0.3834 | **0.4134** |
| | | | Yes | **0.3210** | **0.2794** | **0.3762** | **0.4100** | 0.4104 |
| | | Text2cpts | No | | 0.2376 | **0.2976** | **0.3140** | **0.3230** |
| | | | Yes | **0.2460** | **0.2724** | 0.2968 | 0.3098 | 0.3184 |

Table 12: Cross-modal retrieval in Flickr30K for ConvNeXt-Base aligned to CLIP-ViT-B/32's text encoder for different alignment methods (with and without weight augmentation) and varying training sample sizes. Metrics evaluated are: Image to text Mean Average Precision (mAP), Top-1 Precision (P@1) and Top-5 Precision (P@5) and Text to Image Top-1 Precision (P@1).

| Img. encoder | Metric | Alignment method | +ImgNet 21K | Training set size | | | | |
|---|---|---|---|---|---|---|---|---|
| | | | | 0 | 1K | 5K | 10K | 30K |
| ConvNeXt-Base | i2t mAP | CSA | No | | 0.0037 | 0.4125 | 0.4706 | 0.4972 |
| | | | Yes | **0.3723** | **0.4695** | **0.4948** | **0.5008** | **0.5065** |
| | | MLP | No | | 0.3722 | 0.5137 | 0.5475 | 0.5947 |
| | | | Yes | **0.4422** | **0.4962** | **0.5496** | **0.5772** | **0.6065** |
| | | Text2cpts | No | | 0.4580 | 0.5748 | 0.5912 | 0.6015 |
| | | | Yes | **0.4182** | **0.5333** | **0.5821** | **0.5946** | **0.6025** |
| | i2t P@1 | CSA | No | | 0.0020 | 0.5270 | 0.5730 | 0.6080 |
| | | | Yes | **0.4610** | **0.5860** | **0.6180** | **0.6250** | **0.6270** |
| | | MLP | No | | 0.4520 | 0.6300 | 0.6620 | 0.7350 |
| | | | Yes | **0.5240** | **0.6110** | **0.6800** | **0.7120** | **0.7520** |
| | | Text2cpts | No | | 0.5650 | **0.7130** | 0.7230 | 0.7230 |
| | | | Yes | **0.4710** | **0.6430** | 0.7090 | **0.7260** | **0.7290** |
| | i2t P@5 | CSA | No | | 0.0016 | 0.3656 | 0.4242 | 0.4412 |
| | | | Yes | **0.3352** | **0.4212** | **0.4440** | **0.4522** | **0.4554** |
| | | MLP | No | | 0.3296 | 0.4578 | 0.4892 | 0.5294 |
| | | | Yes | **0.3918** | **0.4434** | **0.4916** | **0.5134** | **0.5424** |
| | | Text2cpts | No | | 0.4060 | 0.5082 | 0.5278 | 0.5396 |
| | | | Yes | **0.3726** | **0.4824** | **0.5164** | **0.5346** | **0.5404** |
| | t2i P@1 | CSA | No | | 0.0014 | 0.1082 | 0.1892 | 0.2342 |
| | | | Yes | **0.1740** | **0.1968** | **0.2512** | **0.2668** | **0.2546** |
| | | MLP | No | | 0.2864 | 0.3796 | **0.4064** | **0.4350** |
| | | | Yes | **0.3594** | **0.3662** | **0.3948** | 0.4050 | 0.4306 |
| | | Text2cpts | No | | 0.2780 | **0.3408** | **0.3512** | **0.3624** |
| | | | Yes | **0.3114** | **0.3082** | 0.3368 | 0.3434 | 0.3592 |

Table 13: Cross-modal retrieval in Flickr30K for TinyViT-21M aligned to CLIP-ViT-B/32's text encoder for different alignment methods (with and without weight augmentation) and varying training sample sizes. Metrics evaluated are: Image to text Mean Average Precision (mAP), Top-1 Precision (P@1) and Top-5 Precision (P@5) and Text to Image Top-1 Precision (P@1).

| Img. encoder | Metric | Alignment method | +ImgNet 21K | Training set size | | | | |
|---|---|---|---|---|---|---|---|---|
| | | | | **0** | **1K** | **5K** | **10K** | **30K** |
| **TinyViT-21M** | i2t mAP | CSA | No | | 0.1501 | 0.4140 | 0.4578 | 0.4853 |
| | | | Yes | **0.3792** | **0.4551** | **0.4807** | **0.4885** | **0.4896** |
| | | MLP | No | | 0.3848 | 0.5032 | 0.5332 | 0.5721 |
| | | | Yes | **0.4375** | **0.4861** | **0.5293** | **0.5459** | **0.5751** |
| | | Text2cpts | No | | 0.4566 | 0.5450 | 0.5580 | 0.5678 |
| | | | Yes | **0.4389** | **0.5145** | **0.5534** | **0.5635** | **0.5709** |
| | i2t P@1 | CSA | No | | 0.1850 | 0.5230 | 0.5840 | **0.6280** |
| | | | Yes | **0.4780** | **0.5710** | **0.6040** | **0.6270** | 0.6200 |
| | | MLP | No | | 0.4780 | 0.6200 | **0.6740** | **0.6990** |
| | | | Yes | **0.5510** | **0.6210** | **0.6610** | 0.6610 | 0.6970 |
| | | Text2cpts | No | | 0.5900 | **0.6790** | 0.6970 | 0.7030 |
| | | | Yes | **0.5380** | **0.6260** | **0.6790** | **0.7010** | **0.7090** |
| | i2t P@5 | CSA | No | | 0.1314 | 0.3714 | 0.4082 | 0.4338 |
| | | | Yes | **0.3438** | **0.4084** | **0.4316** | **0.4414** | **0.4422** |
| | | MLP | No | | 0.3398 | 0.4490 | 0.4738 | 0.5098 |
| | | | Yes | **0.3890** | **0.4292** | **0.4718** | **0.4890** | **0.5154** |
| | | Text2cpts | No | | 0.4012 | 0.4808 | 0.4990 | 0.5058 |
| | | | Yes | **0.3914** | **0.4656** | **0.4958** | **0.5070** | **0.5082** |
| | t2i P@1 | CSA | No | | 0.0168 | 0.1184 | 0.2192 | **0.2404** |
| | | | Yes | **0.1420** | **0.1952** | **0.2330** | **0.2578** | 0.2278 |
| | | MLP | No | | 0.2682 | **0.3586** | **0.3756** | **0.3884** |
| | | | Yes | **0.3808** | **0.3186** | 0.3536 | 0.3646 | 0.3874 |
| | | Text2cpts | No | | 0.2878 | 0.3458 | 0.3574 | 0.3638 |
| | | | Yes | **0.3570** | **0.3584** | **0.3590** | **0.3600** | **0.3662** |

