# OpenReview forum: "Recycling Pretrained Classification Heads for Efficient Vision-Language Alignment"
_ICLR.cc/2026/Conference — Submitted to ICLR 2026_

### Official Review · Reviewer_tGfL · 2025-10-22

**Soundness:** 3
**Presentation:** 3
**Contribution:** 2
**Rating:** 2
**Confidence:** 4

**Summary:**

This paper aims to achieve vision-language alignment of individually trained image and text encoders using a small dataset of image-text pairs.
Instead of collecting a large-scale image-text dataset, the paper proposes to utilize the classification head weights of a pre-trained ImageNet-21K classifier.
Since each row of the weight matrix can be viewed as a prototype of each class, pairs of the weight rows and corresponding class name texts can be used as an additional dataset for training a model (e.g., MLP) that maps text embedding space to image embedding space.
Experimental results demonstrate that the proposed method outperforms baselines with a limited amount of image-text pair data.

**Strengths:**

- S1: Image-text alignment with only a small dataset is helpful for practitioners.
- S2: The proposed method can be easily implemented by simply extracting the head weights of a pre-trained classifier.

**Weaknesses:**

- W1: The motivation for aligning individually trained image and text encoders is ambiguous since many pre-trained CLIP models are available in general classification domains.
- W2: Recent rich unimodal image encoders are mainly trained with self-supervised learning, such as contrastive learning (e.g., SimCLR, MoCo) or masked autoencoders, rather than classification. Can the proposed method also be applied to such models?
- W3: A theoretical explanation of why the classifier weights trained with cross-entropy can be used for prototypes of a cosine classifier would be beneficial.
- W4: A direct evaluation of alignment in the embedding space could be a verification of alignment, which is the primary purpose of the paper. For example, visualizing embedding space and cosine similarity distribution, as is done in [a] and [b], would be helpful.
- W5: The generality of the proposed method is limited since it can achieve competitive accuracy with CLIP only when the domain is close to ImageNet-21K. Can other classification heads trained on other datasets be used?


[a] Eslami and de Melo, Mitigate the Gap: Investigating Approaches for Improving Cross-Modal Alignment in CLIP, ICLR 2025.
[b] Liang et al., Mind the gap: Understanding the modality gap in multi-modal contrastive representation learning, NeurIPS 2022.

**Questions:**

Please refer to the weaknesses.

---

> ### Author Response · Authors · 2025-11-20
>
> We thank the reviewer for the comments. Please find our responses below:
>
> > **W1: The motivation for aligning individually trained image and text encoders is ambiguous since many pre-trained CLIP models are available in general classification domains.**
>
> We acknowledge the widespread availability of generic pre-trained CLIP models. However, the core motivation for post-hoc alignment—a well-established research direction (e.g., ASIF, Text2Concepts, CSA)—is flexibility, modularity. We might require specific vision backbones for particular reasons, such as inference efficiency or model size. Existing CLIP models force users to adopt specific architectures (typically standard ViTs or ResNets). Furthermore, this line of research offers valuable scientific insights into the geometric alignment of independent unimodal representations.
>
> ***
>
> > **W2: Recent rich unimodal image encoders are mainly trained with self-supervised learning, such as contrastive learning (e.g., SimCLR, MoCo) or masked autoencoders, rather than classification. Can the proposed method also be applied to such models?**
>
> Our method leverages classifier weights from labeled training as high-quality representations to augment alignment data. While we acknowledge that recent unimodal encoders are often pre-trained via self-supervised learning, these models can be fine-tuned on labeled datasets to validate features or adapt to downstream tasks. For instance, some checkpoints of architectures like EVA or Swin Transformer (e.g., as found in the `timm` library) have undergone a supervised fine-tuning phase following their self-supervised pre-training. In such cases, our approach can leverage the resulting classification heads to augment alignment data, effectively endowing these image encoders with text capabilities.
>
> ***
>
> > **W3: A theoretical explanation of why the classifier weights trained with cross-entropy can be used for prototypes of a cosine classifier would be beneficial.**
>
> We thank the reviewer for suggesting this theoretical elaboration. In response, we have added Appendix G, which shows why cross-entropy trained weights function as effective prototypes for a cosine classifier.
>
> Our explanation proceeds in two steps:
>
> * **Intuitive View:** We discuss the degrees of freedom, arguing that the vast majority of semantic information is encoded in the direction of the weight vector $w_i$, rather than its magnitude or bias.
> * **Formal Explanation via Neural Collapse:** We leverage the Neural Collapse (NC) framework [1]. We show that under the NC properties of intra-class variability collapse (NC1), convergence to self-duality (NC3) and simplification to nearest class-center (NC4), the standard linear classifier analytically converges to a Cosine Classifier using the weight vectors as prototypes.
>
> Crucially, however, as we clarify in the revised manuscript, the formal theory of Neural Collapse relies on strong assumptions (e.g., balanced data, zero training error, number of classes less than feature dimension). While recent extensions address some limitations individually, gaps remain between these theoretical assumptions and our specific setting. This is why the direct empirical validation provided in Table 2 remains a crucial piece of evidence, demonstrating that the approach holds true in practice even when theoretical conditions are not perfectly met.
>
> ***
>
> > **W4: A direct evaluation of alignment in the embedding space could be a verification of alignment, which is the primary purpose of the paper. For example, visualizing embedding space and cosine similarity distribution, as is done in [a] and [b], would be helpful.**
>
> We appreciate the reviewer’s suggestion to provide visualizations of embeddings and cosine similarity distributions. While we had previously established the existence of a modality gap quantitatively (via permutation tests and MLP discrimination), we agree that direct visualization provides a more intuitive verification of the alignment.
>
> In response, we have added two specific visualizations:
>
> * **Embedding Space Visualization (Figure 6):** We added a UMAP visualization to capture the global structure of the embedding space. As noted in the revised main text, this confirms that while the modalities are semantically aligned, they occupy distinct geometric regions.
> * **Cosine Similarity Distribution (Appendix H):** Following the specific request, we have added a plot showing the distribution of cosine similarities. This verifies that despite the geometric gap, the aligned representations yield significantly higher similarity scores for positive pairs compared to negative ones.
>
> We updated the manuscript to include specific details about the modality gap analysis in the main article (paragraph in Section 4 titled "Modality Gap Between Classification Head Weights and Image Representations").

---

> > ### Author Response · Authors · 2025-11-20
> >
> > ***
> >
> > > **W5: The generality of the proposed method is limited since it can achieve competitive accuracy with CLIP only when the domain is close to ImageNet-21K. Can other classification heads trained on other datasets be used?**
> >
> > We clarify that our method is not strictly bound to the ImageNet domain because it is designed as a complementary augmentation strategy rather than a standalone solution. By adopting a "Union" approach, we combine recycled weights with available image-text pairs. Consequently, our method scales naturally with additional paired data from the target domain while providing a robust performance lower bound in low-data regimes. Regarding the use of other classification heads, heads associated to other datasets can be combined with ImageNet weights as high-quality representations for alignment, provided they share the latent feature space with the former image encoder (e.g., as in a linear probe).
> >
> > **References**
> > [1] Papyan, Vardan, X. Y. Han, and David L. Donoho. "Prevalence of neural collapse during the terminal phase of deep learning training." Proceedings of the National Academy of Sciences 117.40 (2020): 24652-24663.

---

> > > ### Comment · Reviewer_tGfL · 2025-11-25
> > >
> > > I appreciate the authors' response and additional experiments.
> > >
> > > My concerns about W1 and W3 have been resolved.
> > > However, as for the remaining weaknesses, I have questions:
> > >
> > > W2: I recognized that some self-supervised vision models have undergone a supervised fine-tuning phase, but many of the off-the-shelf self-supervised models are not fine-tuned on ImageNet-21k. Thus, the generality is still questionable.
> > > For example, can we obtain ImageNet-21k weights by linear probing or class-mean features?
> > >
> > > W4: I appreciate additional visualizations. But I meant the modality gap between text and image representations. That is, I am curious about how the modality alignment was improved by incorporating ImageNet-21k weights. I am sorry for the ambiguity.
> > >
> > > W5: I understand that the proposed method can be incorporated with available image-text pairs. However, the effect of the difference between the target and ImageNet-21k domains is unexplored. For example, is the proposed method effective in specialized domains, such as medical domains?

---

> > > > ### Author Response · Authors · 2025-11-26
> > > >
> > > > We thank the reviewer for the follow-up comments. Please find our responses below:
> > > >
> > > > > **W2: I recognized that some self-supervised vision models have undergone a supervised fine-tuning phase, but many of the off-the-shelf self-supervised models are not fine-tuned on ImageNet-21k. Thus, the generality is still questionable. For example, can we obtain ImageNet-21k weights by linear probing or class-mean features?**
> > > >
> > > > Indeed, as the reviewer suggests, linear probing weights and class-mean features can effectively serve as representations for post-hoc alignment. Our proposal specifically highlights ImageNet-21K pre-training weights because they are often an available by-product of the training process, providing high-quality representations without the need for additional computation. However, if linear probes are available (e.g., from standard evaluation protocols) or if class-mean features have been computed, these can certainly be "recycled" in the exact same manner to provide robust alignment signals.
> > > >
> > > > ***
> > > >
> > > > > **W4: I appreciate additional visualizations. But I meant the modality gap between text and image representations. That is, I am curious about how the modality alignment was improved by incorporating ImageNet-21k weights. I am sorry for the ambiguity.**
> > > >
> > > > We appreciate the clarification, in response, we included Figure 10 and added the following paragraph to Appendix E:
> > > >
> > > > *"Visualizing the Impact of Weight Recycling on Alignment. To visually investigate how incorporating ImageNet-21K weights affects the geometric configuration of embeddings from an MLP-aligned model, we visualize the latent space geometry using UMAP. Figure 10 displays the embeddings of CIFAR-100 class texts ($\textup{MLP}(f_{T}(\text{'A photo of } <class \ i>'))$) and the average image representations of the corresponding classes (via the image encoder $f_I$). As shown in the random alignment baseline (Fig. 10a), the representations present no structure; specifically, a text representation of a class is not closer to its corresponding image representation than to any other image representation. When training the alignment MLP using ImageNet-1K weights (Fig. 10b), we observe an organization in the spatial distribution. We note that this configuration results in a downstream accuracy in zero-shot classification in CIFAR-100 of less than 60% (see Fig. 3). Crucially, when training with ImageNet-21K weights (Fig. 10c), we observe a significantly more structured configuration. In this case, the downstream accuracy rises to 73.66% (see Table 7 and Figure 3), demonstrating that the richer semantic information in the 21K weights leads to better alignment in this downstream task."*
> > > >
> > > > ***
> > > >
> > > > > **W5: I understand that the proposed method can be incorporated with available image-text pairs. However, the effect of the difference between the target and ImageNet-21k domains is unexplored. For example, is the proposed method effective in specialized domains, such as medical domains?**
> > > >
> > > > We thank the reviewer for the question. To assess the effectiveness of our method in a domain significantly different from ImageNet-21K, we conducted an additional zero-shot evaluation on the HAM10000 dataset (dermatological skin lesion image classification). We observed performance levels comparable to those of CLIP. Specifically, we included Table 9 and added the following paragraph in Appendix E:
> > > >
> > > > *"Zero-shot classification accuracy in a specialized domain dataset. We evaluate the zero-shot performance of our post-hoc alignment on a specialized domain exhibiting a significant distribution shift from ImageNet-21K. We employ the HAM10000 dataset […], a benchmark for dermatological skin lesion image classification comprising 7 classes. Table 9 compares a post-hoc aligned model (BEiT-B16 aligned via recycled ImageNet-21K weights) against ViT CLIP baselines and a random classification baseline. We report Balanced Accuracy to account for class imbalance. The results indicate that the specialized nature of dermatological images presents a challenge for all evaluated Vision-Language Models. Both CLIP and our proposed method achieve performance only marginally above the random baseline (14.28%). In particular, our method achieves a balanced accuracy of 18.32%, similar to the CLIP ViT-B/32 variant (18.85%). This suggests that the alignment learned from ImageNet-21K weights is as robust as CLIP's text-image alignment when applied to unseen images in specialized domains. In addition, these results confirm that for such specific tasks, fine-tuning is necessary for any general-purpose VLM. This fine-tuning can be realized either end-to-end like CLIP or in a post-hoc manner as followed in our approach."*

---

### Official Review · Reviewer_zgtA · 2025-10-24

**Soundness:** 3
**Presentation:** 3
**Contribution:** 3
**Rating:** 4
**Confidence:** 4

**Summary:**

This paper proposes recycling classification head weights from ImageNet-21K pretraining as data augmentation to achieve data-efficient vision-language alignment. Experiments on cross-modal retrieval, zero- and few-shot classification tasks demonstrate the effectiveness of proposed method.

**Strengths:**

1. This paper proposes a simple-yet-effective way for data-efficient post-hoc vision-language alignment, which recycles classification head weights from ImageNet-21K pretraining as augmented training data.
2. This paper also provides analysis on why we can treat pretrained classification head weights as image representations
3. Experiments on several downstream tasks show the effectiveness of proposed method.

**Weaknesses:**

1. The authors only highlight the improvements of their method over existing post-hoc alignment techniques such as CSA, Text-to-Concept, and MLP alignment in the context of cross-modal retrieval. However, it is also essential to validate these gains on zero- and few-shot classification tasks, as such comparisons are fundamental to demonstrating its overall effectiveness. Furthermore, it is recommended that these results be included as a table in the main paper.

2. The authors are encouraged to provide t-SNE visualizations in the embedded space, comparing the original image/text representations with the classification head weights. Such visualization would help reveal the potential modality gap and offer a more comprehensive understanding of the alignment effect.

3. Regarding the third zero-shot configuration (ImageNet-21K & 1 Image-Caption pair per class), the inclusion of one image from each class in the evaluation datasets raises the question of whether this setting constitutes a strictly zero-shot scenario.

4. The proposed method appears to bind the classification head weights to a specific image encoder. For instance, when using head weights derived from ViT-B/16 pre-training, the image encoder must also be the pre-trained ViT-B/16. If this understanding is correct, it would limit the flexibility and general applicability of the approach. Besides, the pretrained classification head weights only contain 21K samples, which may cause scalability concerns.

5. Can the pretrained classification head weights be leveraged to enhance a pre-trained CLIP model? Specifically, can we fine-tune adapters on the 21K data samples derived from these weights to improve CLIP performance?

Minor Issue:
In Line 304-305, $g$ and $\overline{g}$ should be exchanged. $g$ is the identity and $\overline{g}$ is the MLP.

**Questions:**

See Weaknesses.

---

> ### Author Response · Authors · 2025-11-20
>
> We thank the reviewer for the comments. Please find our responses below:
>
> > **Validation on Classification Tasks:** The authors only highlight the improvements of their method over existing post-hoc alignment techniques such as CSA, Text-to-Concept, and MLP alignment in the context of cross-modal retrieval. However, it is also essential to validate these gains on zero- and few-shot classification tasks, as such comparisons are fundamental to demonstrating its overall effectiveness. Furthermore, it is recommended that these results be included as a table in the main paper.
>
> We agree on the importance of validating gains across classification tasks. Following the reviewer's suggestion to evaluate all methods' gains on zero-shot classification, we have included Figure 4 in the main text to highlight the MLP results, while detailing the consistent gains for CSA and Text2Concepts in Appendix E (Fig. 7) due to space constraints.
>
> Regarding few-shot classification, we restricted our evaluation to the MLP method because our approach employs a sequential training strategy (pre-training on weights, then fine-tuning on few-shot samples). This strategy specifically leverages the parametric nature of the MLP, whereas closed-form methods like CSA do not naturally support such sequential gradient-based fine-tuning.
>
> ***
>
> > **Visualization of Modality Gap:** The authors are encouraged to provide t-SNE visualizations in the embedded space, comparing the original image/text representations with the classification head weights. Such visualization would help reveal the potential modality gap and offer a more comprehensive understanding of it.
>
> We thank the reviewer for this suggestion. We have included these visualizations in Appendix E (Figure 6). We specifically opted for UMAP (Uniform Manifold Approximation and Projection) rather than t-SNE, as UMAP is known to better preserve the global structure of the data [1]. This property makes it particularly well-suited for visualizing the macro-level geometric separation between the image representations and the classification weights. The resulting plots clearly visualize the distributions of original image features versus classification weights, confirming that both modalities occupy distinct regions of the latent space.
>
> ***
>
> > **Zero-shot Validity:** Regarding the third zero-shot configuration (ImageNet-21K & 1 Image-Caption pair per class), the inclusion of one image from each class in the evaluation datasets raises the question of whether this setting constitutes a strictly zero-shot scenario.
>
> To contextualize our approach, we note that even large-scale VLMs like CLIP exhibit significant semantic overlap between pre-training data and downstream benchmarks. Xu et al. [2] reconstruct CLIP’s data curation process and report that over 700 of the 1,000 ImageNet classes appear in the pre-training metadata, observing a positive correlation between this overlap and downstream zero-shot performance. This suggests that semantic overlap between pre-training concepts and downstream tasks is a common factor in the success of state-of-the-art VLMs.
>
> Accordingly, we consider the use of ImageNet-21k pre-training weight representations—and the associated single image-caption pair per class—to be consistent with the "zero-shot" terminology as established in the literature. As we infer from Xu et al. (2024), our usage of a single pair per class is likely far less than the number of images from downstream tasks CLIP encountered during pre-training. In addition, we emphasize that our main aim is to demonstrate how image-caption data can be effectively combined with weight representations to increase downstream performance. This experiment specifically highlights that even a minimal signal (one pair per class) combined with recycled weights yields robust alignment.
>
> Lastly, to clarify this concern, we have added the following clarification to the revised manuscript: *"Xu et al. (2024) reconstructs CLIP’s data curation process and finds over 700 out of the 1K classes in ImageNet-1K present in pretraining metadata and observes a correlation between downstream zero-shot classification accuracy and the number of classes matched in the metadata. We can also infer from Xu et al. (2024) that our use of 1 image-caption pair per class across all datasets is likely less than the number of overlapping image-caption pairs CLIP saw during its pretraining phase."*

---

> > ### Author Response · Authors · 2025-11-20
> >
> > ***
> >
> > > **Flexibility and Scalability:** The proposed method appears to bind the classification head weights to a specific image encoder. For instance, when using head weights derived from ViT-B/16 pre-training, the image encoder must also be the pre-trained ViT-B/16. If this understanding is correct, it would limit the flexibility and general applicability of the approach. Besides, the pretrained classification head weights only contain 21K samples, which may cause scalability concerns.
> >
> > Regarding the claim that binding weights to the image encoder limits flexibility, we argue that this correspondence is inherent to the usage of pre-trained models, not a constraint. Our method applies to any image encoder (e.g., ResNet, ViT, ConvNeXt) pre-trained on large-scale labeled datasets, where the corresponding classification heads are readily available as standard artifacts of pre-training. There is no practical motivation to apply mismatched weights (e.g., ViT weights to a ResNet encoder) when the native, high-quality semantic representations are inherently provided for each specific image backbone pre-trained on a large-scale labeled dataset.
> >
> > Regarding scalability, we emphasize that our method is a data augmentation technique, not a standalone algorithm limited to 21k samples. We employ a "Union" strategy (Eq. 3 from the paper) that combines recycled weights with any volume of available image-text pairs, and we demonstrate the effectiveness of this combination in Figure 3. Consequently, our method is not bounded by the number of classes; it scales naturally with additional paired data while providing competitive performances in low-data regimes.
> >
> > ***
> >
> > > **Application to CLIP:** Can the pretrained classification head weights be leveraged to enhance a pre-trained CLIP model? Specifically, can we fine-tune adapters on the 21K data samples derived from these weights to improve CLIP performance?
> >
> > Yes, this is a promising direction. Specifically, we envision this being particularly valuable in scenarios where a CLIP image encoder is fine-tuned on a large-scale classification dataset (e.g., ImageNet-21K or iNaturalist). In such cases, the original vision-language alignment is often compromised due to the shift in the visual embedding space. In this context, the classification head weights $w_i$ generated during fine-tuning serve as updated prototypes of the visual space. We can effectively use the pairs $(w_i, t_i)$—where $t_i$ is the class description—as augmentation data for the alignment dataset. The underlying principle of recycling weights as semantic anchors is equally applicable to augment alignment data for the full/adapter fine-tuning of text encoders, as long as the image encoder is kept frozen so that classification weights live in the same feature space as the one produced by the image encoder.
> >
> > ***
> >
> > > **Minor Issue:** In Line 304-305.
> >
> > We appreciate the correction and have fixed the notation swap in the revised manuscript.
> >
> > **References**
> >
> > [1] McInnes, L., Healy, J., & Melville, J. (2018). UMAP: Uniform Manifold Approximation and Projection for Dimension Reduction. arXiv preprint arXiv:1802.03426.
> >
> > [2] Hu Xu et al. Demystifying clip data. International Conference on Learning Representations (ICLR) 2024.

---

### Official Review · Reviewer_WDG4 · 2025-10-30

**Soundness:** 3
**Presentation:** 2
**Contribution:** 3
**Rating:** 4
**Confidence:** 3

**Summary:**

This paper proposes a pragmatic, data-efficient method to perform post-hoc alignment between independently pretrained image and text encoders. The key idea is to recycle the row vectors (class-weight vectors) from classification heads learned during ImageNet-21K pretraining and pair each weight vector wi with a text embedding of the corresponding class name fT(ti). These (wi, fT(ti)) pairs are used as additional training data (Dweights) and combined with a (potentially small) set of image–text pairs (Dimgtxt) to learn lightweight alignment mappings (g, g) with existing post-hoc methods (e.g., CSA, text-to-concepts, or a small MLP).

**Strengths:**

1. Recycling classification head weights is simple, computationally cheap, and leverages resources (pretrained weights) that are commonly discarded.
2. The authors test multiple alignment methods (CSA, text-to-concepts, MLP) and several vision encoders, evaluate on retrieval (Flickr30K) and a diverse set of nine classification benchmarks, compare ImageNet-1K vs ImageNet-21K weights, and present ablations (adding one image-caption per class).

**Weaknesses:**

1.The authors observe a significant geometric separation between weight vectors and image features (an "image-weight modality gap") but proceed by simply concatenating both modalities into the same alignment dataset. There is little exploration of principled ways to handle this gap (e.g., modality-specific normalizations, learned projections for weight vectors, or reweighting strategies).
2. The approach requires access to the pretraining classifier weights and associated human-readable class names. Many commercial or proprietary vision models do not expose their classification-layer weights or class-label vocabulary.

**Questions:**

1. You report a statistically significant separation between weight vectors and image features. Have you tried learning a dedicated lightweight projection (or simple normalization / scaling) for the weight vectors prior to alignment, or adding modality-specific alignment heads? If so, does that improve retrieval and classification compared to the current union strategy? If not, can you comment on the expected benefit and computational cost?
2. Your strongest results rely on ImageNet-21K classification heads. For target domains with little or no overlap with ImageNet concepts (e.g., medical imaging, non-photographic domains), what is the recommended strategy? Can you (or will you) provide experiments using alternative large-scale label sets (e.g., Places365, iNaturalist) or show how selecting a subset of ImageNet-21K weights relevant to the target domain affects performance?

---

> ### Author Response · Authors · 2025-11-20
>
> We thank the reviewer for the comments. Please find our responses below:
>
> > **Geometric separation and gap mitigation:** The authors observe a significant geometric separation between weight vectors and image features (an "image-weight modality gap") but proceed by simply concatenating both modalities into the same alignment dataset. There is little exploration of principled ways to handle this gap (e.g., modality-specific normalizations, learned projections for weight vectors, or reweighting strategies).
> >
> > *(And associated question 1)* You report a statistically significant separation between weight vectors and image features. Have you tried learning a dedicated lightweight projection (or simple normalization / scaling) for the weight vectors prior to alignment, or adding modality-specific alignment heads? If so, does that improve retrieval and classification compared to the current union strategy? If not, can you comment on the expected benefit and computational cost?
>
> We appreciate the suggestion to explore principled ways to handle the "image-weight modality gap." We clarify that we had initially explored a statistical mitigation strategy consisting of centering and rescaling the classifier's weights $w_{i}$ via the transformation
>  $$w_{i} \rightarrow \frac{w_{i} - \mu_{w} + \mu_{x}}{||w_{i} - \mu_{w} + \mu_{x}||}$$
>  where $\mu_{w}$ and $\mu_{x}$ represent the global means of the weights and image representations, respectively, which remain fixed for every weight row transformed. We thus, chose to stick to the simpler "Union" baseline (concatenation) because the previous preprocessing did not yield performance improvements.
>
> Following the reviewer’s specific suggestion, we further explored a learned lightweight projection approach. We found that this parametric strategy did not improve the simple concatenation baseline. We have added these details to Appendix H (in the paragraph "Testing modality gap mitigation strategies" and Fig. 9). These results are consistent with those given by Liang et al. [1], who also observe that closing the geometric modality gap does not necessarily guarantee better downstream alignment. As already noted in our Limitations section, we leave the exploration of how more advanced mitigation techniques might help to improve downstream performance for future work.
>
> ***
>
> > **Dependence on internal weights:** The approach requires access to the pretraining classifier weights and associated human-readable class names. Many commercial or proprietary vision models do not expose their classification-layer weights or class-label vocabulary.
>
> We acknowledge that our method relies on accessing model weights, making it not suitable for black-box APIs. We target open-weight models pre-trained on large-scale labeled datasets like ImageNet-21k. In this standard setting (e.g., via `timm` library), the dense classification weights and vocabularies are publicly available resources. Our approach is designed to efficiently "recycle" the rich semantic representations provided by classifier weight vectors to endow an image encoder with text capabilities.

---

> > ### Author Response · Authors · 2025-11-20
> >
> > ***
> >
> > > **Domain specificity:** Your strongest results rely on ImageNet-21K classification heads. For target domains with little or no overlap with ImageNet concepts (e.g., medical imaging, non-photographic domains), what is the recommended strategy? Can you (or will you) provide experiments using alternative large-scale label sets (e.g., Places365, iNaturalist) or show how selecting a subset of ImageNet-21K weights relevant to the target domain affects performance?
> >
> > We appreciate the reviewer's suggestion to explore how our method behaves with alternative large-scale label sets, particularly for domain-specific applications. In response, we have added Appendix I (**"Alignment using classifier weights from a domain-specific dataset"**), where we evaluate our method using classification weights from a ConvNeXt model pre-trained on iNaturalist—a large-scale dataset focused on fine-grained flora and fauna—rather than ImageNet-21K.
> >
> > Our results indicate that using ImageNet-21K weights generally yields superior downstream performance, likely due to the greater semantic diversity of its classes which forces the alignment to cover a broader region of the shared latent space. While the iNaturalist-aligned models exhibit a clear domain preference—achieving relatively better results on biological datasets like Flowers102, OxfordPets, and CIFAR—their performance drops significantly on datasets lacking specific plant or animal content, confirming that semantic overlap is a critical factor. Based on these findings, our recommended strategy is to leverage the broadest available classifier (like ImageNet-21K) for general-purpose tasks to maximize semantic coverage. However, for highly distinct target domains (e.g., medical imaging), the "recycling" principle remains applicable: if a robust classifier pre-trained on that specific domain is available, its weights can still be leveraged to augment aligning data.
> >
> > **References**
> > [1] Liang, Weixin, et al. "Mind the gap: Understanding the modality gap in multi-modal contrastive representation learning." NeurIPS (2022).

---

### Official Review · Reviewer_1e3R · 2025-11-04

**Soundness:** 2
**Presentation:** 2
**Contribution:** 2
**Rating:** 4
**Confidence:** 4

**Summary:**

The paper presents a data-efficient post-alignment method for aligning vision and language representations. Building on post-hoc alignment techniques, it reuses classification head weights from ImageNet21K pretraining as a form of data augmentation， by treating the head weight as concept representation, combining them with a smaller set of image–text pairs. This approach reduces training cost while maintaining strong performance. When integrated with existing post-hoc alignment methods, it consistently improves results on cross-modal retrieval and zero- or few-shot classification, offering a versatile and efficient alternative to full-scale end-to-end vision–language training.

**Strengths:**

The idea of enhancing post-hoc alignment by averaging multiple image representations per class to form a prototype is novel and appealing. It improves efficiency by consolidating representations, though it sacrifices some diversity in the text–image relationship. As a result, the model becomes primarily classification-oriented, losing finer details such as compositional or structural information present in the original images and text.

**Weaknesses:**

Insufficient related work discussion: Lines 136–161 largely replicate content from [1], yet fail to include a proper comparison or discussion, even though [1] demonstrates stronger performance than the chosen baseline.


Contradictory motivation: The motivation emphasizes using minimal paired data for post-hoc alignment, but the main experiments rely on a CLIP text encoder trained on massive paired datasets—undermining the stated goal.


Experimental issues:


- The first experiment trains and tests on Flickr30k, which fails to test generalization—the key strength of CLIP. The improvement over the baseline is partly due to adding 21k new classes, making comparisons unfair. A fair baseline should use 21k ImageNet images with captions and apply the vanilla post-hoc alignment method.


- The second experiment aligns the text encoder to a classification head and evaluates it on overlapping datasets with ImageNet21k, essentially exploiting the evaluation setup. This converts a strong non–zero-shot model into a pseudo–zero-shot one rather than demonstrating true zero-shot ability.


Overall, the experiments lack evidence of generalization or transfer learning benefits, which are central to CLIP’s purpose.

[1] Zhang, Le, Qian Yang, and Aishwarya Agrawal. "Assessing and Learning Alignment of Unimodal Vision and Language Models." Proceedings of the Computer Vision and Pattern Recognition Conference. 2025.

**Questions:**

how would the model really transfer to downstream task, say coco retrieval or other tasks such as MMVP, Winoground, which tests fine-grained and compositional understanding of the model beyond simple classificaiton?

---

> ### Author Response · Authors · 2025-11-20
>
> We thank the reviewer for the comments. Please find our responses below:
>
> > **Insufficient related work discussion:** Lines 136–161 largely replicate content from [1], yet fail to include a proper comparison or discussion, even though [1] demonstrates stronger performance than the chosen baseline.
>
> The formulation in lines 136–161 serves to define the general problem setting under which various cited methodologies in the literature (e.g., CSA, ASIF, Text2Concepts) operate. We acknowledge that [1] constitutes an additional method addressing this same general problem—specifically through linear mappings and contrastive loss—and thus fits within the scope of this unified framework. Our intention was to provide a broad description that encompasses the diverse works operating in this space.
>
> However, regarding our experimental comparison, we prioritized baselines capable of operating effectively in limited-data regimes. While [1] is relevant to the problem, it necessitates large-scale data for alignment, utilizing the CC3M dataset (approximately 2.2M pairs) augmented with LLM-generated captions. In contrast, our work focuses on data efficiency, leveraging 21k weight representations and significantly fewer paired samples (less than 1k for zero-shot classification and ~10k for retrieval). We have revised the manuscript to include a reference to [1], explicitly noting its reliance on millions of paired samples: *"SAIL (Zhang et al., 2025) contrastively trains two linear transformations on millions of image-caption pairs to align representations from both modalities."*
>
> ***
>
> > **Contradictory motivation:** The motivation emphasizes using minimal paired data for post-hoc alignment, but the main experiments rely on a CLIP text encoder trained on massive paired datasets—undermining the stated goal.
>
> Our objective is specifically to mitigate the data requirements for the alignment step and demonstrate how a "by-product" of ImageNet-21k pre-training—the classification weights—can be recycled as high-quality representations for vision-language alignment. We clarify that while the CLIP text encoder is pre-trained on massive data, we treat it as a fixed, off-the-shelf component. We are endowing an image encoder with text capabilities using minimal or no paired data—a standard setup in post-hoc alignment (e.g., Text2Concepts, CSA). Furthermore, our Appendix experiments with text-only encoders (RoBERTa, MPNet) confirm our method works effectively even without CLIP’s inherent paired-data priors.
>
> To make this distinction explicit, we have added the following clarification to the manuscript: *"We note that the goal of our post-hoc alignment method is to minimize the paired data required for the alignment step itself by leveraging high-quality representations provided by (unimodal) classifier weights. This objective is distinct from the data requirements of the pre-training phase for the image and text encoder components, which we treat as fixed, off-the-shelf models."*
>
> ***
>
> > **The first experiment trains and tests on Flickr30k, which fails to test generalization—the key strength of CLIP.**
>
> We test generalization in Figure 2 (corresponding to 0 training samples), where our method achieves effective cross-modal retrieval on Flickr30K without using any image-caption pairs to train the alignment. This result is difficult to attribute to memorization of ImageNet categories alone, since Flickr30K queries contain complex descriptions rather than single-label matches. In addition, and following the reviewer’s suggestion (in the question), we have included MS-COCO zero-shot retrieval experiments in Table 3. These results show that leveraging recycled classification weights for alignment yields non-trivial retrieval performance, further reinforcing that the model generalizes beyond simple class-level memorization.

---

> ### Author Response · Authors · 2025-11-20
>
> ***
>
> > **The improvement over the baseline is partly due to adding 21k new classes, making comparisons unfair. A fair baseline should use 21k ImageNet images with captions and apply the vanilla post-hoc alignment method.**
>
> We clarify that the classification weights are an inherent artifact of the pre-training process, incurring no extra data collection cost. However, following the reviewer's valuable suggestion, we conducted an ablation study (Appendix D, Tab. 5) comparing alignment using these weights versus an equivalent number of ImageNet image-text pairs. The results show that recycled weights significantly outperform the corresponding image-text pairs in downstream retrieval. This confirms that the performance gain also derives from the superior semantic quality of the weight representations, not merely the quantity of alignment points.
>
> To reflect this insight in the paper, we added a clarification to the cross-modal retrieval section: *"A key question is whether improved retrieval stems merely from additional alignment data. We first note our method uses classification weight representations at no extra cost—they are pre-training by-products requiring no new image-text pairs. Secondly, we conducted an ablation study (Appendix D) comparing downstream retrieval performance when alignment uses weight representations versus an equivalent number of image-text pairs. Results show classification weights outperform image-text pairs, despite the latter matching the downstream task's data modality. This highlights the superior quality of reusing weight representations for alignment, extending their utility beyond their original classification single purpose."*
>
> ***
>
> > **The second experiment aligns the text encoder to a classification head and evaluates it on overlapping datasets with ImageNet21k, essentially exploiting the evaluation setup. This converts a strong non–zero-shot model into a pseudo–zero-shot one rather than demonstrating true zero-shot ability.**
>
> We appreciate the reviewer’s concern about potential “pseudo-zero-shot” evaluation and have taken actions to clarify it in our methodology. First, to place our approach in context we note that prior work has shown large-scale VLMs pretraining datasets can contain semantically overlapping classes from downstream benchmarks: In [2], they reconstruct CLIP’s data curation and report that over 700 of the 1k ImageNet classes appear in pretraining metadata, with a positive correlation between downstream zero-shot classification accuracy and the number of matched classes in the metadata. This means semantic overlapping between pretraining data and downstream tasks also happens in CLIP (it actually might be the reason for CLIP’ success [2]). Accordingly, using ImageNet-21k pretraining weights representations is consistent with the “zero-shot” terminology as used in the literature.
>
> To directly address the reviewer’s concern, we have added explicit clarification along these lines in the revised manuscript: *"While ImageNet-21k pretraining weight representations may contain semantic overlap with downstream benchmarks, we note that CLIP has also likely seen most of the classes in the downstream classification benchmarks (Xu et al., 2024). For instance, Xu et al. (2024) reconstructs CLIP’s data curation process and finds over 700 out of the 1K classes in ImageNet-1K present in pretraining metadata and observes a correlation between downstream zero-shot classification accuracy and the number of classes matched in the metadata"*
>
> ***
>
> > **How would the model really transfer to downstream task, say coco retrieval or other tasks such as MMVP, Winoground, which tests fine-grained and compositional understanding of the model beyond simple classification?**
>
> We included extended experiments on COCO zero-shot retrieval (Table 3). However, we do not evaluate on MMVP or Winoground since our objective is to demonstrate the efficiency of recycling classification weights for standard alignment tasks. Solving the compositionality and fine-grained understanding typically requires massive training (e.g. as in [1]), which contradicts our low-resource, post-hoc motivation.
>
> **References**
>
> [1] Zhang, Le, Qian Yang, and Aishwarya Agrawal. "Assessing and Learning Alignment of Unimodal Vision and Language Models." Proceedings of the Computer Vision and Pattern Recognition Conference. 2025
>
> [2] Hu Xu et al. Demystifying clip data. International Conference on Learning Representations (ICLR) 2024.

---

### Author Response · Authors · 2025-11-20
**General Response**

We thank the reviewers for their constructive feedback. We revised the paper to address their comments, with key changes summarized below:

### 1. New Experiments & Ablations

* **MS-COCO Retrieval (Tab. 3):** Added to demonstrate generalization beyond class memorization **(Reviewer 1e3R)**.
* **Zero-Shot Classification:** We included new experiments on zero-shot classification, placing the MLP results in the main text (Fig. 4) and other methods (CSA, Text2Concepts) in Appendix E **(Reviewer zgtA)**.
* **Superior quality of weight representations (Appendix D):** Added an ablation showing that recycled classification weights significantly outperform an equivalent number of image-text pairs, hence proving the improvement does not merely stem from the addition of new representations but from the superior quality of representations provided by weights **(Reviewer 1e3R)**.
* **Domain Specificity (Appendix I):** Added analysis using iNaturalist weights, showing how domain-specific classifiers affect fine-grained alignment **(Reviewer WDG4)**.

### 2. Theoretical & Geometric Analysis

* **Theoretical Grounding (Appendix Theory):** Added an explanation based on Neural Collapse to explain why cross-entropy weights act as effective prototypes **(Reviewer tGfL)**.
* **Visualization (Fig. 6 & Appendix H):** Added UMAP plots and cosine similarity distributions to visualize the geometry of the modality gap **(Reviewers zgtA, tGfL)**.
* **Gap Mitigation (Appendix H):** Provided experiments on how basic modality gap mitigation techniques impact downstream performance **(Reviewer WDG4)**.

Below we provide detailed answers tackling each of the comments/issues raised by the reviewers.

---

### Meta-Review · Area_Chair_97gs · 2025-12-10

**Summary:**

Four knowledgeable reviewers assessed this submission. The reviewers found the problem relevant (tGfL),  the idea novel (1e3R) and simple (WDG4, zgtA, tGfL) but were not convinced by the presented results (1e3R, WDG4)

Their main concerns can be summarized as:
1. Missing comparisons/positioning with related work (1e3R), improvements beyond cross-modal retrieval (zgtA), image backbones trained with ssl (tGfL)
2. Unclear motivation (1e3R, tGfL)
3. Experimental validation appeared unconvincing (generalization, zero-shot setups, domain specificity, fairness of comparisons) (1e3R, WDG4, zgtA, tGfL), and requirement to accessing pretrained classifier weights and associated class names (WDG4)
4. Limited exploration of principled ways to handle the image-weight modality gap (WDG4)
5. Unclear generalizability, scalability, and applicability of the approach between different architectures (zgtA, tGfL)
6. Missing theoretical explanation of why classifier weights trained with cross-entropy may be used as prototypes (tGfL)

**Reviewer Concerns:**

The authors' rebuttal partially addresses the reviewers' concerns. In particular, the response argues that CLIP is treated as a fixed, off-the-shelf component and that the work endows an image encoder with text capability using limited amounts of paired data. Authors note the consistent use of "zero-shot' terminology across the literature. The rebuttal includes additional experiments on MS-COCO (zero-shot), ablation studies, classification weights from a model trained on iNaturalist, HAM10000 dataset (with performance levels similar to CLIP), and distribution and embedding visualizations requested by the reviewers. Finally, the authors clarify that there is no practical motivation to apply mismatched weights, that they are not bound to the ImageNet domain, nor bounded by the number of classes and that the method scales naturally with additional paired data points.

However, the rebuttal only argues against the inclusion of some baselines and datasets (e.g., Winoground which has shown to be challenging for VL-based approaches) and does not experimentally address the concern related to the direct use of SSL models (possibly with linear probes). Adding those would make the experimental validation more convincing and the significance of the presented results clearer.

**Reviewer Scores:**

Given that the rebuttal only partially addressed the reviewers' concerns, the AC thinks the scores would have likely remained borderline.

---

### Decision · Program_Chairs · 2026-01-26

Reject